# Orient Anything: Learning Robust Object Orientation Estimation from Rendering 3D Models

**Zehan Wang** [1][†]  **Ziang Zhang** [1][†]  **Tianyu Pang** [2][♠]  **Chao Du** [2]  **Hengshuang Zhao** [3]  **Zhou Zhao** [1][*]

https://orient-anything.github.io/

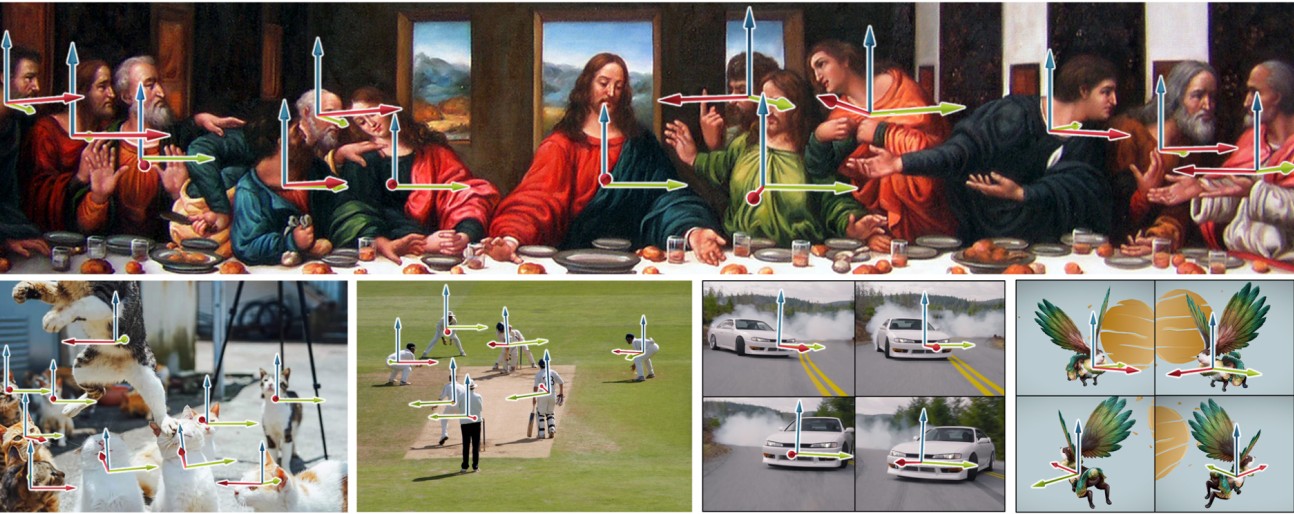

*Figure 1.* We introduce a novel method for estimating the object orientation in images, represented by the red axis, while the blue and green axes indicate the upward and left sides of the object. More examples are provided in Appendix. *Best viewed on screen with zoom.*

## Abstract

Orientation is a fundamental attribute of objects, essential for understanding their spatial pose and arrangement. However, practical solutions for estimating the orientation of open-world objects in monocular images remain underexplored. In this work, we introduce **Orient Anything**, the first foundation model for zero-shot object orientation estimation. A key challenge in this task is the scarcity of orientation annotations for open-world objects. To address this, we propose leveraging the vast resources of 3D models. By developing a pipeline to annotate the front face of 3D objects and render them from random viewpoints, we curate 2 million images with precise orientation annotations across a wide variety of object cate-

gories. To fully leverage the dataset, we design a robust training objective that models the 3D orientation as probability distributions over three angles and predicts the object orientation by fitting these distributions. Besides, we propose several strategies to further enhance the synthetic-to-real transfer. Our model achieves state-of-the-art orientation estimation accuracy on both rendered and real images, demonstrating impressive zero-shot capabilities across various scenarios (Fig. 1). Furthermore, it shows great potential in enhancing high-level applications, such as understanding complex spatial concepts in images and adjusting 3D object pose.

## 1. Introduction

Perceiving object properties in a single image is the core problem in computer vision. Current visual foundation models and large vision-language models (VLMs) excel in tasks like object recognition (Zhang et al., 2024; Liu et al., 2024), localization (Liu et al., 2023; Li et al., 2022), tracking (Wang et al., 2023; Rajič et al., 2023), and segmen-

---

[†]Equal contribution ,[♠]Project leader ,[*]Corresponding author ,
[1]Zhejiang University [2]Sea AI Lab [3]The University of Hong Kong.
Correspondence to: Zhou Zhao <zhaozhou@zju.edu.cn>.

*Proceedings of the 42nd International Conference on Machine Learning*, Vancouver, Canada. PMLR 267, 2025. Copyright 2025 by the author(s).

tation (Kirillov et al., 2023; Ravi et al., 2024). However, estimating orientation of open-world objects, which is critical for understanding object pose and arrangement, has been underexplored due to the lack of annotated data. ObjectNet3D (Xiang et al., 2016) and Omni3D (Brazil et al., 2023) enable 3D orientation prediction with manually annotated objects, but its scope is still restricted to specific domains and limited object categories, making it difficult to generalize to diverse real-world scenarios.

Furthermore, even the most advanced general visual understanding systems, such as GPT-4 (Hurst et al., 2024) and Gemini (Team et al., 2023; 2024), struggle to comprehend basic object orientation. The perception of this fundamental object attribute has not emerged in current vision-language models. As a result, they perform poorly on tasks involving orientation, such as predicting object movement trends or understanding spatial relationships, as shown in Fig. 2.

In this paper, we first propose to learn how various objects look under different orientations by rendering 3D models. By annotating the front face of massive 3D objects (Deitke et al., 2023), we can easily and cheaply obtain precise orientation labels for rendered views. In particular, we leverage advanced VLM (Team et al., 2024) to identify the front side of 3D objects from orthographic views, complemented by canonical pose detection and symmetry analysis to simplify the task and improve accuracy. Then, we render images from random perspectives, using azimuth and polar angles relative to the 3D orientation vector, combined with the camera rotation angle, to represent the 3D orientation. This idea provides scalable, diverse, and easy-to-acquired data, enabling the development of accurate and generalizable orientation estimation models.

Although scalable orientation data is available now, training a reliable orientation prediction model remains non-trivial. Direct regression of the three angles struggles to converge, resulting in poor performance. To overcome this challenge, we reformulate the single angle values as probability distributions to better capture the correlation between adjacent angles. By driving the model to fit these angle probability distributions, we simplify the learning process and significantly enhance model robustness. Furthermore, considering the domain gap between the rendered and real images, we investigate various model initializations to better incorporate real-world prior knowledge, alongside data augmentation strategies to improve synthetic-to-real transfer.

Our contribution can be summarized as:

- We develop an automatic and reliable 3D object orientation annotation pipeline, and highlight the values of rendering 3D objects for cost-effective, diverse, and scalable images with precise orientation labels.

- We introduce the orientation probability distribution

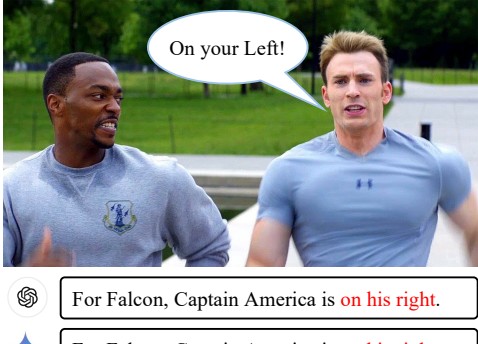

From Falcon's (left man in image) perspective, is Captain America (right man in image) on his left or right?

Figure 2. Understanding object orientation is essential for spatial reasoning. However, even advanced VLMs like GPT-4o and Gemini-1.5-pro are not yet able to resolve the basic orientation issue.

fitting task as the learning objective to stabilize the training process and improve generalization.

- We investigate various model initialization and data augmentation strategies to improve synthetic-to-real transfer.

- Our model exhibits much stronger orientation estimation ability compared to both the expertise model (Cube RCNN) and leading VLMs (GPT-4o and Gemini).

## 2. Related Work

### 2.1. Object Orientation Estimation in Images

Some tasks attempt to recognize object orientations in images under certain conditions or with extra information.

Object viewpoint estimation (Su et al., 2015; Tulsiani & Malik, 2015) aims to predict the object viewpoints for limited categories. 6DoF object pose estimation (Guan et al., 2024) focuses on detecting the 3D position and orientation of objects in images. However, these methods typically require the guidance of CAD model or other reference views of the target object (Sundermeyer et al., 2018; Peng et al., 2019; Goodwin et al., 2022; Nguyen et al., 2024; Fan et al., 2024), and are often limited to a small set of categories.

Rotated object detection (Xie et al., 2021; Han et al., 2021; Brazil et al., 2023) is concerned with generating rotatable 2D or 3D bounding boxes for objects in images. Omni3D (Brazil et al., 2023) unifies multiple 3D object detection datasets and develops Cube R-CNN. While Cube R-CNN demonstrates certain abilities in detecting object orientation in 3D space, it is primarily trained on indoor scenes and street environments, limiting its zero-shot generalization to in-the-wild scenarios.

Unlike the aforementioned tasks, our work focuses on 3D orientation estimation of open-world objects and only requires monocular images.

## 2.2. Orientation-based Understanding

Object orientation provides context about how objects are positioned relative to one another and to the viewer (the camera), which is fundamental for object pose and relationship understanding. Accurate orientation understanding plays a key role in many advanced applications.

In 3D scene understanding, many studies (Chen et al., 2020; Achlioptas et al., 2020; Azuma et al., 2022) have highlighted the importance of spatial relationships informed by object orientation. SQA3D (Ma et al., 2022) first describes the position and orientation of an agent in a 3D scene, then tasks the model with answering questions based on the given spatial context. EmboidedScan (Wang et al., 2024) manually annotates orientations for 3D objects and utilizes the pose information to describe spatial relationships among objects in 3D space.

In the domain of 2D images, understanding object orientation is also fundamental for accurately interpreting (Góral et al., 2024; Wu et al., 2024) or generating (Shi et al., 2023; Huang et al., 2024; Wei et al., 2023) spatial relationships and properties. Góral et al. (2024) propose the visual perspective-taking task to assess 2D VLM's ability to understand the orientation and viewpoint of a person in images and highlight various applications based on this ability. Furthermore, the object orientation relative to the camera determines its pose in the image, which is essential for distinguishing spatial properties such as the front wheels of cars and the left shoulder of a person, along with complex spatial relationships. Moreover, generating objects with given pose conditions is vital for controllable image generation (Huang et al., 2024; Wei et al., 2023).

Although object orientation is related to numerous applications, practical solutions for estimating the orientation of arbitrary objects in images are still underexplored. Our work fills this gap by proposing the first foundation model for object orientation estimation, which exhibits strong zero-shot performance in real-world scenarios.

## 3. Orientation Understanding in 2D VLMs

Before proposing our method for object orientation estimation, we need to address an important question: "*Do current 2D VLMs, trained on web-scale image datasets with billions of parameters, inherently learn to distinguish the orientation of various objects?*"

To this end, we introduce Ori-Bench, the first VQA benchmark specifically designed to assess the capacity of 2D

| | Object Direction | Spatial Part | Spatial Relation | Overall |
|---|---|---|---|---|
| Random | 12.93 | 22.12 | 17.54 | 16.75 |
| GPT-4o | 49.32 | 15.38 | 27.27 | 32.50 |
| Gemini-1.5-pro | 58.90 | 15.38 | 18.18 | 33.00 |
| Orient Anything+LLM | **67.12** | **46.15** | **40.91** | **51.50** |

*Table 1.* Quantitative results on the proposed Ori-Bench.

VLMs to understand object orientation and tackle related questions. We manually curate 200 images in total, with 100 from COCO (Lin et al., 2014) and 100 generated by DALL-E 3 (Betker et al., 2023). To substantively evaluate the understanding of object orientation, each image is horizontally flipped to produce a paired mirrored version, with answers adapted accordingly. A sample will be marked as solved only if the model correctly answers the question on both versions. There are three kinds of tasks: *(1) Object Direction Recognition (73+73 samples):* identifying the orientation of an object within images; *(2) Spatial Part Reasoning (39+39 samples):* distinguish parts of an object with specific spatial meanings, like left *vs.* right hand of human; and *(3) Spatial Relation Reasoning (88+88 samples):* imagining the relative position of one object from the perspective of another.

In Tab. 1, we show the accuracy of GPT-4o and Gemini-1.5-Pro. In the basic direction recognition task, the advanced VLMs are only able to correctly solve around 60% of the samples. This limitation further impacts their performance in spatial reasoning and relation questions, where the powerful GPT-4o and Gemini-1.5-Pro perform similarly to random guessing. The pilot study highlights the need for fundamental tools to precisely estimate object orientation in images. Our simple baseline, Orient Anything+LLM (see Sec. 7.1 for details), outperforms the powerful GPT-4o and Gemini-1.5-Pro in these orientation-related tasks.

## 4. Orientation Data Collection

The scarcity of annotations is the main obstacle to learning general orientation estimation. Existing annotations for images, typically captions (Schuhmann et al., 2021; 2022), bounding boxes (Lin et al., 2014), or segmentation masks (Kirillov et al., 2023; Zhou et al., 2017), seldom include object orientation information, and manually annotating object orientation in images is extremely time-consuming and costly. To overcome this limitation, we propose to utilize 3D assets. Annotating the front face of 3D objects and rendering images from random perspectives provides an efficient and effective way to generate large-scale images with precise orientation annotations.

To this end, we first develop an automatic 3D asset annotation and rendering pipeline, as shown in Fig. 3. Each step of the pipeline is detailed below.

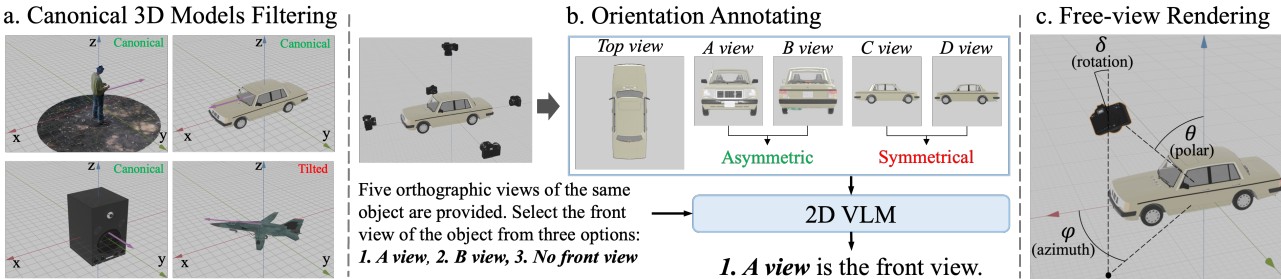

*Figure 3.* The orientation data collection pipeline is composed of three steps: **1) Canonical 3D Model Filtering**: This step removes any 3D objects in tilted poses. **2) Orientation Annotating**: An advanced 2D VLM is used to identify the front face from multiple orthogonal perspectives, with view symmetry employed to narrow the potential choices. **3) Free-view Rendering**: Rendering images from random and free viewpoints, and the object orientation is represented by the polar $\theta$, azimuthal $\varphi$ and rotation angle $\delta$ of the camera.

**Step1: Canonical 3D Models Filtering** We use Objaverse (Deitke et al., 2023), a large-scale dataset containing 800K object assets, as our database. Although most objects in this dataset are modeled in canonical poses[1], some are tilted along orthogonal axes, as shown in Fig. 3.a. To simplify the annotation process and enhance reliability, we first exclude all the tilted assets, focusing solely on 3D objects in canonical poses. This idea reduces the 3D object orientation annotation to a classification task. Rather than identifying the specific orientation vector, we only need to determine the front face from images rendered alone $x$, $-x$, $y$, and $-y$ axes, or to conclude that the object has no front face.

To filter out tilted objects, we analyze the tilt along the $x$, $y$, and $z$ axes for each object. Specifically, we extract the object edges for each view and use Principal Component Analysis (PCA) to identify the principal directions of these edges. If the principal edge direction is aligned with any coordinate axis (with a tolerance of $2°$ for robustness) across all renderings, the object is considered to be in the canonical pose. Otherwise, it is deemed tilted.

Starting with the initial pool of 800K objects in the Objaverse dataset, we first curate 80K 3D models with high-quality texture. Applying our tilt-filtering criteria, we select 55K objects in canonical poses for subsequent processing.

**Step2: Orientation Annotating** Using the selected 3D objects in canonical poses, we render five orthogonal views from the $x$, $-x$, $y$, $-y$ and $z$ axes. Although our pilot study in Sec. 3 indicates that current 2D VLMs struggle to accurately predict orientation from a single view, we find that they perform well in identifying which view is facing the camera when multiple orthogonal views are presented for comparison and reference, as shown in Fig. 3.b.

Additionally, to mitigate VLM hallucinations and improve annotation accuracy, we incorporate symmetry as auxiliary

information. Since the front and back faces of objects are typically asymmetrical, we leverage this prior knowledge to further narrow down the possible choices. Specifically, we use a combination of SIFT (Lowe, 2004), structural similarity, and pixel color similarity to assess the similarity between opposing views. Two views are considered symmetrical if their similarity exceeds the threshold. Gemini-1.5-Pro is tasked with identifying the front face of objects from asymmetrical opposing views. If the object is symmetrical along both the $x$ vs. $-x$ and $y$ vs. $-y$, it is regarded as having no meaningful front face and orientation.

**Step3: Free-View Rendering** Once the 3D asset's front face is annotated in 3D space, we can obtain its 3D orientation in images from any viewpoints. For simplicity and clarity, we use the spherical coordinate system to define object orientation. As depicted in Fig. 3.c, we calculate the relative polar angle $\theta$ and azimuth angle $\varphi$ between the camera position and the object orientation axis, as well as the camera rotation angle $\delta$, to represent the object orientation from the specific viewpoint.

Before rendering, all 3D objects are scaled to a unit cube, with their centers aligned to the origin of the coordinate system. For each object, 40 images are rendered from random perspectives, with the camera aimed at the origin and each image rendered at $512 \times 512$ resolution. In total, we collect 2M rendered images with precise orientation annotations.

**Statistics** To assess the diversity of our dataset, we evaluate its categorical coverage across WordNet entities (Fellbaum, 2010), following the methodology of Objaverse (Deitke et al., 2023). The 55K objects we used span 7,204 distinct entities. In contrast, the previous 3D dataset with orientation annotations, ObjectNet3D (Xiang et al., 2016), includes only 100 categories.

---

[1]In our definition, standing upright and facing one of four orthogonal directions along the $x$, $-x$, $y$, and $-y$ axes

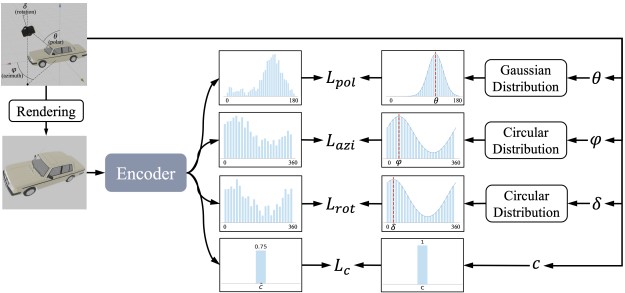

*Figure 4.* Orient Anything consists of a simple visual encoder and multiple prediction heads. It is trained to judge if the object in the input image has a meaningful front face and fits the probability distribution of 3D orientation.

## 5. Orient Anything

Based on the massive images of objects with annotated 3D orientation $\theta$, $\varphi$, and $\delta$, we train Orient Anything for general object orientation estimation in images.

### 5.1. Orientation Probability Distribution Fitting

Despite the availability of precise 3D orientation annotations, formulating an effective learning objective for robust orientation estimation remains a non-trivial challenge. The initial approach, which involved directly predicting continuous angle values with MSE loss as supervision, faced convergence issues and yielded suboptimal performance.

To address this, we first simplify the challenging continuous regression task into a discrete classification problem, which is easier to optimize. Specifically, we divide the 360° range into 360 individual classes, each representing a 1° interval. While lowering the task difficulty improves performance over continuous regression, it fails to capture correlations between adjacent angles that produce nearly identical outcomes in practice (e.g., rendering at polar 29°, 30°, and 31°). Treating these close angles as independent classes neglects their inherent relationships, which may confuse the model. Therefore, we further reformulate the classification task as a discrete probability distribution fitting problem, which is also easy to converge and can fully capture the potential relationship between different orientations.

**Target Probability Distribution**  We first transform the ground-truth angles into target probability distributions, represented as Gaussian distributions centered on the ground-truth angle, with manually set variances. These distributions are subsequently discretized into a grid-based format at 1° intervals. For a given ground-truth polar angle $\theta$ (in degrees), the probability distribution of polar angle $\mathbf{P}_{\text{pol}}(i|\theta, \sigma_\theta)$ can

be formulated as follows:

$$\mathbf{P}_{\text{pol}}\left(i|\theta, \sigma_\theta\right) = \frac{\exp\left(-\frac{(i-\theta)^2}{2\sigma_\theta^2}\right)}{\sum_{n=1}^{180} \exp\left(-\frac{(n-\theta)^2}{2\sigma_\theta^2}\right)}, \quad (1)$$

where $i = 1°, \ldots, 180°$ and $\sigma_\theta$ is the variance hyperparameter for polar distribution. For the ground truth azimuth angle $\varphi$ and rotation angle $\gamma$, due to its periodicity (e.g., 359°, 360°, and 1° are adjacent), we employ the circular Gaussian distribution to form their target distribution $\mathbf{P}_{\text{azi}}(i|\varphi, \sigma_\varphi)$ and $\mathbf{P}_{\text{rot}}(i|\delta, \sigma_\delta)$. For brevity, we illustrate this process using azimuth as an example:

$$\mathbf{P}_{\text{azi}}\left(i|\varphi, \sigma_\varphi\right) = \frac{\exp\left(\frac{\cos(i-\varphi)}{\sigma_\varphi^2}\right)}{2\pi I_0\left(\frac{1}{\sigma_\varphi^2}\right)}, \quad (2)$$

where $i = 1°, \ldots, 360°$, $\sigma_\varphi$ is the variance for polar distribution, and $I_0(\frac{1}{\sigma_\varphi^2})$ is the zero-order modified Bessel function of the first kind, which can be represented as:

$$I_0\left(\frac{1}{\sigma_\varphi^2}\right) = \sum_{n=0}^{\infty} \frac{1}{(n!)^2}\left(\frac{1}{2\sigma_\varphi^2}\right)^{2n}. \quad (3)$$

As shown in Fig. 4, the circular Gaussian distribution effectively models the periodicity of azimuth and rotation angles, ensuring the stability of the optimization process.

**Training and Inference**  Given the input image $I$, we use a visual encoder to extract its latent feature, followed by prediction heads (simple linear layers) to output the distributions of polar, azimuth and rotation angles: $\widehat{\mathbf{P}}_{\text{pol}} \in \mathbb{R}^{180}$, $\widehat{\mathbf{P}}_{\text{azi}} \in \mathbb{R}^{360}$, and $\widehat{\mathbf{P}}_{\text{rot}} \in \mathbb{R}^{360}$, respectively, representing the object orientation in 3D space. Additionally, the model outputs an orientation confidence $\hat{\mathbf{c}} \in \mathbb{R}^1$, to indicate whether the object has a meaningful front face and to aid in identifying centrally symmetric objects, such as balls and stools.

The target distributions $\mathbf{P}_{\text{pol}}(i|\theta, \sigma_\theta)$, $\mathbf{P}_{\text{azi}}(i|\varphi, \sigma_\varphi)$ and $\mathbf{P}_{\text{rot}}(i|\delta, \sigma_\delta)$ are defined above, with the orientation label $\mathbf{c}$ being 1 if the object has a front face, and 0 otherwise. We use cross-entropy (CE) loss to supervise the predicted orientation distributions, and the corresponding loss terms are denoted as: $L_{\text{pol}}$, $L_{\text{azi}}$ and $L_{\text{rot}}$. For $\hat{\mathbf{c}}$, binary cross-entropy (BCE) loss is employed, yielding $L_{\text{c}}$. The final training loss is a linear combination of the above four terms, and for objects without meaningful orientation, the $L_{\text{pol}}$, $L_{\text{azi}}$, $L_{\text{rot}}$ will be disabled:

$$L = \begin{cases} \lambda L_{\text{c}}, & \mathbf{c} = 0 \\ L_{\text{pol}} + L_{\text{azi}} + L_{\text{rot}} + \lambda L_{\text{c}}, & \mathbf{c} = 1 \end{cases} \quad (4)$$

where $\lambda$ is the loss coefficient for orientation judgment.

During the inference process, objects whose orientation confidence is lower than 0.5 would be thought to have no meaningful front face and orientation. Otherwise, the angles with the highest probability in each distribution: $\widehat{\mathbf{P}}_{\text{pol}}$, $\widehat{\mathbf{P}}_{\text{azi}}$, $\widehat{\mathbf{P}}_{\text{rot}}$ are taken as the predicted polar, azimuth, and rotation angle: $\hat{\theta}$, $\hat{\varphi}$, $\hat{\delta}$.

### 5.2. Sythetic-to-Real Transferring

Although the rendered images of 3D objects provide extensive data with orientation annotations, there is a distribution shift between synthetic rendered images and real images. We try to prompt effective synthetic-to-real transfer from two aspects: integrating real-world pre-training knowledge and narrowing the training-inference domain gap.

**Inheriting Real-world Knowledge by Initialization** As demonstrated in (Yang et al., 2024; Ke et al., 2024), initializing the model with strong visual encoders pre-trained on real images can significantly improve its synthetic-to-real transfer ability. To evaluate this in our orientation estimation task, we train models initialized from 3 widely-used image pre-trained encoders: MAE (He et al., 2022), CLIP (Radford et al., 2021), and DINOv2 (Oquab et al., 2023). After trials and failures, DINOv2 yields satisfactory results, attributed to its task-agnostic pre-training, fine-grained perception, and strong generalization capabilities. Consequently, we develop our model using DINOv2 initialization.

**Narrowing Domain Gap by Data Augmentation** There are two main differences between rendered and real images. We employ corresponding data augmentation strategies to reduce the domain gap and enhance transfer performance.

First, objects in rendered images are typically fully visible, whereas real-world images often contain partially visible or occluded objects. To bridge this gap, we incorporate random cropping as a training data augmentation strategy. This technique simulates the occlusion situation in real-world images, thereby improving the model's ability to generalize to real-world scenarios.

Second, to avoid ambiguity, the rendered image contains only a single object, whereas real-world images often feature multiple objects. Given the remarkable success of current object detection and segmentation models, our approach only focuses on unambiguous orientation estimation for a single object. To extend our model to multi-instance scenarios, we will isolate each object using a segmentation model and estimate objects' orientation individually. This strategy not only enhances the model's applicability to real-world images but also mirrors the style of rendered images.

## 6. Experiments

### 6.1. Implementation Details

We train models at three scales for different purposes: ViT-S, ViT-B, and ViT-L, all initialized with DINOv2. The loss coefficient $\lambda$ in Eq. 4 is set to 1. The variance hyperparameters $\sigma_\theta$, $\sigma_\varphi$, and $\sigma_\delta$ are configured as 2.0°, 20.0°, and 1.0°. For optimization, we use the AdamW (Loshchilov, 2017) optimizer, with a learning rate of 1e-5 for the pre-trained visual encoder and 1e-3 for the newly introduced prediction heads. The models are trained for 50,000 steps with a batch size of 64 on the curated 2M object orientation dataset. All trainings are conducted on 4 A100 (40GB) GPUs.

### 6.2. Zero-shot Real-world Orientation Recognition

The primary goal of this work is to estimate orientations of open-world objects in real images. To assess the model performance in real-world scenarios, we construct two kinds of evaluation benchmarks.

1) For objects in domains previously explored, we compare Orient Anything with Cube RCNN(a fully-supervised model) (Brazil et al., 2023) across four indoor and street datasets: SUN RGB-D (Song et al., 2015), KITTI (Geiger et al., 2012), nuScenes (Caesar et al., 2020), and Objectron (Ahmadyan et al., 2021). For each dataset, 1,000 objects with 3D orientation annotations were randomly selected and cropped from real images. We separately compute the Absolute Error (Abs) for azimuth, polar, and rotation angles for comparison. Additionally, we benchmark our model against the zero-shot and fine-tuned baselines on ImageNet3D (Ma et al., 2024), currently the largest dataset for object orientation annotations. We also compare Orient Anything with FSDetView(a few-shot model) (Xiao et al., 2022) on Pix3D (Sun et al., 2018) and Pascal3D+ (Xiang et al., 2014). Acc@30° and Abs on the overall 3D spherical space are reported, following FSDetView.

2) For objects in the wild, we collect samples from COCO (Lin et al., 2014) and manually annotate their orientations. Due to the challenges of accurately annotating 3D orientations, we simplify the task by labeling object orientations on the horizontal plane in eight directions: front, back, left, right, front-left, front-right, back-left, and back-right. From the 80 categories in the COCO validation set, we select 20 images per category, resulting in a comprehensive benchmark of 1,600 samples. Two tasks are used for evaluation: *(1) Orientation Judgment:* Determine whether the object has a meaningful orientation; *(2) Horizontal Orientation Recognition:* Identify the direction the object is facing on the horizontal plane, selecting from one of the eight pre-defined directions.

As shown in Tabs. 2, 3 and 4, despite never being exposed to real-world images during training, each version of Orient

| Models | SUN RGB-D | | | KITTI | | | nuScenes | | | Objectron | | | Pascal3D+ | Pix3D | |
|---|---|---|---|---|---|---|---|---|---|---|---|---|---|---|---|
| | Azimuth Abs↓ | Polar Abs↓ | Rotation Abs↓ | Azimuth Abs↓ | Polar Abs↓ | Rotation Abs↓ | Azimuth Abs↓ | Polar Abs↓ | Rotation Abs↓ | Azimuth Abs↓ | Polar Abs↓ | Rotation Abs↓ | Acc@30°↑ | Abs↓ | Acc@30°↑ |
| FSDetView† | - | - | - | - | - | - | - | - | - | - | - | - | 46.00 | 38.3 | 49.00 |
| Cube RCNN‡ | 93.58 | 39.73 | 140.10 | 98.61 | 39.73 | 121.21 | 89.63 | 15.64 | 132.57 | 122.99 | 60.01 | 113.31 | - | - | - |
| Ours (ViT-S) | 58.20 | 11.63 | **3.59** | 65.85 | 5.00 | 1.08 | 72.68 | 5.58 | 2.16 | 39.45 | 23.47 | 18.26 | 46.00 | 37.50 | 68.00 |
| Ours (ViT-B) | 56.34 | 9.15 | 3.75 | 54.02 | 5.86 | **0.21** | 66.56 | 5.72 | **1.28** | 36.49 | **22.13** | 18.34 | 48.00 | 35.17 | **69.00** |
| Ours (ViT-L) | **42.98** | **8.38** | 3.66 | **44.22** | **3.57** | 0.89 | **55.17** | **4.08** | 1.78 | **30.09** | 22.19 | 18.54 | **55.00** | **22.90** | 66.00 |

*Table 2.* Zero-shot orientation estimation on six unseen real-image benchmarks. † indicates that the model is used in a few-shot manner, while ‡ denotes fully-supervised models. Orient Anything, on the other hand, is used in a zero-shot manner. These "**unequal**" comparisons further emphasize the effectiveness of our models. The best results are highlighted in **bold**.

| Setting | Models | Avg. | Electronics | Furniture | Household | Music | Sports | Vehicles | Work |
|---|---|---|---|---|---|---|---|---|---|
| Zero-shot | ImageNet3D-ResNet50 | 37.1 | 30.1 | 35.6 | 28.1 | 11.8 | **51.7** | 36.7 | **40.9** |
| | Ours (ViT-B) | **48.5** | **61.0** | **66.8** | **37.9** | 27.3 | 25.6 | **70.8** | 33.4 |
| Fine-tuning | ImageNet3D-ResNet50 | 53.6 | 49.2 | 52.4 | 45.8 | 26.0 | 65.2 | 56.5 | 58.5 |
| | ImageNet3D-DINOv2-B | 64.0 | 75.3 | 47.9 | 32.9 | 23.5 | **74.7** | 38.1 | **64.0** |
| | Ours (ViT-B) | **71.3** | **77.6** | **89.7** | **64.4** | **54.4** | 47.6 | **87.4** | 61.2 |

*Table 3.* Orientation estimation on Large-scale ImageNet3D, which covers 200 object categories. The Acc@30° results are reported.

| Models | Rendered Image | | | | Real Image | |
|---|---|---|---|---|---|---|
| | Judgment Acc↑ | Azimuth Acc@22.5°↑ | Polar Acc@5°↑ | Rotation Acc@5°↑ | Judgment Acc↑ | Recognition Acc↑ |
| Random | 50.00 | 12.50 | 5.55 | 16.67 | 50.00 | 12.50 |
| Cube RCNN | - | 12.44 | 10.37 | 2.50 | - | 20.25 |
| Gemini-1.5-pro | 57.29 | 19.06 | 16.31 | 85.12 | 66.96 | 31.95 |
| GPT-4o | 61.85 | 19.94 | 17.56 | 81.00 | 69.29 | 45.78 |
| Ours (ViT-S) | 73.88 | 63.18 | 71.62 | 97.06 | 78.54 | 63.44 |
| Ours (ViT-B) | 74.88 | 71.94 | 81.37 | **99.56** | **81.25** | 70.19 |
| Ours (ViT-L) | **76.00** | **73.94** | **86.75** | 98.31 | 80.30 | **72.44** |

*Table 4.* Orientation estimation on both in-domain rendered images and out-of-domain real images. The best results are **bold**.

| Models | Single View | Canonical Views | Canonical& Symmetrical |
|---|---|---|---|
| Gemini-1.5-pro | 44.00 | 74.00 | **86.00** |
| GPT-4o | 31.00 | 87.00 | **92.00** |

*Table 5.* Ablation study for Orientation Annotation.

Anything clearly outperforms existing methods in recognizing object orientations in real images. Orient Anything consistently exceeds the performance of previous approaches by a significant margin across most categories, achieving over 70% accuracy in the majority of COCO categories. Detailed results on each benchmark are provided in Appendix.

We visualize our model predictions in Fig. 1 and 7, 6, 8-12 in Appendix. These qualitative results highlight Orient Anything's remarkable zero-shot capability across images captured or created by real cameras, human artists, or generative models, as well as a variety of scenarios, including continuous video frames, multi-view images, and complex scenes containing multiple objects.

### 6.3. Rendered-Images Orientation Estimation

We first quantitatively validate models by estimating the numerical 3D orientation of the in-domain rendered images. We manually select and annotate 300 objects from Objaverse, including 150 with orientation annotations and 150 without a meaningful orientation. For each object, we render 16 images from random viewpoints, resulting in a total of 4,800 images for evaluation.

We evaluate methods from two aspects: *1) Orientation Judgment*: Determine if the object has a meaningful front face. *2) Separated Orientation Estimation*: Predict the accurate azimuth, polar, and camera rotation angles for objects, using Acc@X° (accuracy within tolerances of $\pm X°$) as metrics. Rotated object detection model: Cube RCNN (Brazil et al., 2023) and VLMs: GPT-4o (Hurst et al., 2024) and Gemini-1.5-pro (Team et al., 2024), are used as baselines.

The results presented in Tab. 4 demonstrate the superior performance of our model in predicting 3D orientation for in-domain rendered images. In practical azimuth estimation, our method achieves more than three times the accuracy of alternatives. Notably, the performance of Cube RCNN and advanced VLMs is only marginally better than random guessing, with a success rate of 19.94%, compared to 12.50%. In contrast, the Orient Anything ViT-L achieves 73.94% accuracy, highlighting its practical value in reliably distinguishing object orientation on the horizontal plane.

Due to similar definitions and the same random guess results, "Acc@22.5°" for rendered image azimuth estimation and "Acc" for real image horizontal direction recognition are comparable. Our models achieve similar results on both metrics, highlighting the excellent synthetic-to-real transfer performance. For VLMs, recognizing horizontal directions in words is more accurate than predicting precise azimuth

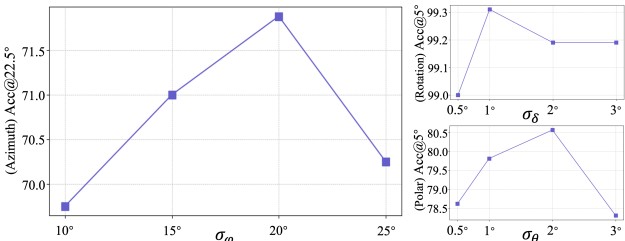

*Figure 5*. Ablation study for hyper-parameter $\sigma_\theta$, $\sigma_\varphi$ and $\sigma_\delta$.

values in numbers, which reveals the shortcomings of VLMs in predicting precise values for 3D orientation. On the other hand, Cube RCNN performs significantly worse on rendered images due to its limited generalization capability.

## 6.4. Ablation Study

To verify the effectiveness of our key designs, we conduct ablation experiments using the ViT-B encoder.

**Designs in Orientation Annotating** We evaluate our orientation annotating methods using the 300 manually annotated 3D objects introduced in Sec. 6.3. The results in Tab. 5 indicate that while VLMs achieve only 44% and 31% accuracy when identifying object orientation from the top view alone, providing orthogonal perspectives substantially enhances performance. Furthermore, incorporating symmetry as an extra condition further raises accuracy to nearly 90%, underscoring the effectiveness of our orientation annotating strategy and proving the reliability of the rendering data.

**Effect of $\sigma_\theta$, $\sigma_\varphi$ and $\sigma_\delta$** Fig. 5 shows the effect of variance hyper-parameter for three kinds of angle probability distribution. In general, our method is insensitive to the variance selection, while most configurations yield superior results compared to the one-shot label.

**Effect of Probability Prediction** In Tab. 6, we ablate the three learning objectives discussed in Sec. 5.1: continuous value regression, discrete angle classification, and probability distribution fitting. Direct regression yields poor performance, while angle classification performs significantly better but remains suboptimal. The final proposed probability distribution fitting method surpasses the alternatives, achieving markedly superior performance.

**Number of Rendering Views** We explore the effect of the number of images rendered pre-3D object in Tab. 6. For a fair comparison, we train models to converge for each setting. The results indicate that too few views fail to provide sufficient information about objects from different perspectives, while overly dense sampling results in redundant images within the dataset, potentially hindering convergence. Empirically, rendering 40 views for each object achieves the best balance and yields the optimal results.

| Design | Variants | Rendering Image | | Real Image |
| | | *Azimuth* | *Polar* | *Recognition* |
| | | Acc@22.5° | Acc@5° | Acc |
| --- | --- | --- | --- | --- |
| Learning Objective | Regression | 12.00 | 20.50 | 21.48 |
| | Classification | 68.75 | 79.00 | 66.93 |
| | Fitting | **71.88** | **80.56** | **69.85** |
| Number of Views | 10 | 67.19 | 78.19 | 63.67 |
| | 20 | 67.94 | 78.88 | 65.47 |
| | 30 | 70.06 | 78.13 | 68.62 |
| | 40 | **71.88** | 80.56 | **69.85** |
| | 80 | 69.12 | **80.69** | 66.48 |
| Training Initialization | CLIP | 58.44 | 71.88 | 49.27 |
| | MAE | 58.44 | 64.63 | 57.26 |
| | DINOv2 | **71.88** | **80.56** | **69.85** |
| Training Augmentation | None | 71.88 | 80.56 | 69.85 |
| | Cropping | **71.94** | **81.37** | **70.19** |
| Inference Augmentation | Box | 71.88 | 80.56 | 67.49 |
| | Mask | 71.88 | 80.56 | **69.85** |

*Table 6*. Ablation study for Learning Objective, Number of Views, Training Initialization and Data Augmentation.

**Effect of Model Initialization** We compare several powerful pre-trained visual encoders as initialization for our orientation estimation task in Tab. 6. We empirically find that DIONv2 exhibits much better performance in both in-domain convergence and out-of-domain transfer compared to others, which may be attributed to its large-scale task-agnostic pre-training and superior fine-grained perception.

**Effect of Data Augmentation** Tab. 6 present the effect of data augmentation for improving sythetic-to-real transfer. During training, random cropping enables rendered images to mimic the object occlusions, which significantly enhances the performance in real-world scenarios. For inference, using segmentation masks to isolate objects aligns more closely with the style of rendered images compared to bounding boxes, thereby narrowing the domain gap and improving overall performance.

## 7. Applications

### 7.1. Spatial Understanding

Orientation is a key attribution for accurately understanding the spatial relations, as we highlighted in Sec. 3 and Fig. 2. We find that using Grounded-SAM (Ren et al., 2024) and our Orient Anything to identify object position and orientation in images, and then conveying these spatial details in pure text to an LLM (Hurst et al., 2024), effectively addresses orientation-based questions that confuse GPT-4o and Gemini-1.5-pro, as shown in Tab. 1 and Fig. 13-15. These results underscore the importance of our model in

enhancing spatial understanding.

## 7.2. Spatial Generation Scoring

As shown in Fig. 7, we empirically find that even leading image generation models, like DALL-E 3 (Betker et al., 2023) and FLUX (Labs, 2024), struggle to generate content that conforms to given object orientation or spatial relationship conditions. Our model can help distinguish whether the generated image follows the given spatial condition, demonstrating its potential as a reward model to guide generative models in adhering to the desired orientation- and perspective-based spatial concepts.

## 7.3. 3D Models Orientation Voting

Many existing 3D data exhibit varied orientations, with some even tilted relative to the coordinate axes. As shown in Fig. 1, our method achieves consistent orientation predictions across multi-view images, enabling robust voting for 3D assets' orientation. Accurately estimating the orientation of 3D assets is valuable for further scaling up rendering images with orientation annotations or adjusting the poses of 3D assets to a desired direction.

## 8. Conclusion

In this paper, we present Orient Anything, a practical approach for estimating the orientation of open-world objects from single images. We design an automated and reliable 3D object annotation and rendering pipeline, enabling the collection of large-scale images with accurate orientation annotations. To fully leverage the value of the new dataset, we design an orientation probability distribution fitting task to learn robust orientation estimation. Additionally, we improve synthetic-to-real transfer performance by incorporating real-world knowledge and minimizing the domain gap. As a result, Orient Anything achieves impressive zero-shot object orientation estimation in real-world images and demonstrates great potential as a foundational tool for enabling applications such as complex spatial understanding and generation scoring.

## Impact Statement

Orient Anything provides a fundamental tool for estimating the orientation of open-world objects in images. This is highly beneficial for high-dimensional applications such as controllable 3D/2D content generation, 3D scene understanding, and autonomous driving. In different application scenarios, it is essential to follow the corresponding usage guidelines to ensure its proper and ethical application, minimizing any potential risks.

## Acknowledgements

This work was supported in part by National Natural Science Foundation of China under Grant No.624B2128, No.62222211 and No.U24A20326

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

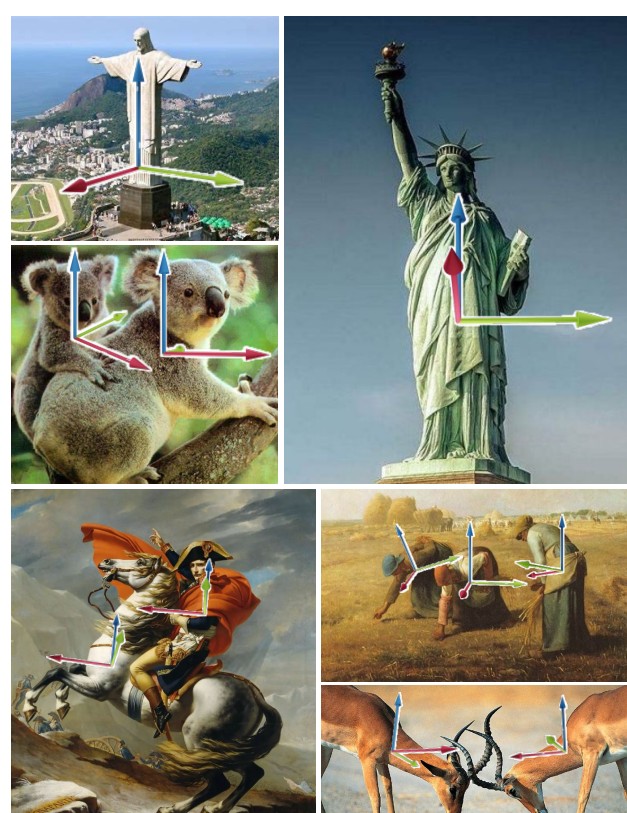

*Figure 6.* More visualization of images in the wild.

## A. Detailed Results on COCO Benchmark

In Tab. 7, we provide the detailed horizontal direction recognition accuracy for each object category in COCO that is annotated with front face and orientation.

Our model achieves excellent performance across most object categories with clear orientations, attaining an accuracy exceeding 80%. However, it performs relatively poorly in categories where the distinction between front and back is ambiguous or the objects are too small. Compared to previous alternatives, Orient Anything achieves significantly better accuracy in most categories than the best results achieved by previous models.

## B. More Visualizations of Images in The Wild

In Fig. 6, we present more visualizations of images from various domains containing different objects. In these images, our model shows consistently accurate orientation prediction results, further highlighting the impressive zero-shot capability of our Orient Anything.

## C. Visualization of Real-image Benchmarks

In Fig. 8, 9, 10, 11 and 12, we present the qualitative results on objects of SUN RGB-D (Song et al., 2015),

*a). "A car is driving **away at 30 degrees to the left.**"* | *b). "Charmander holding ice cream **in its left hand.**"*

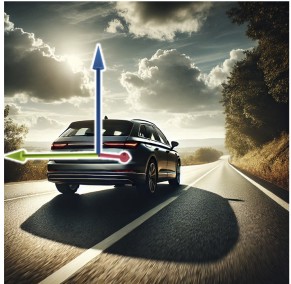 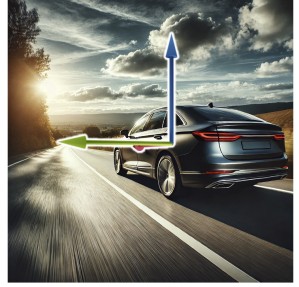 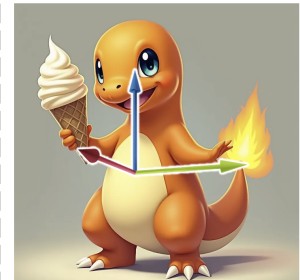 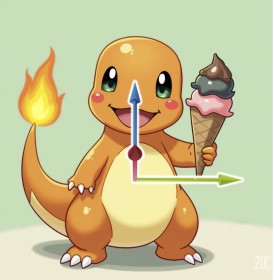

Case 1 (30° to the right) | Case 2 (30° to the left) | Case 1 (in the right hand) | Case 2 (in the left hand)

*Figure 7.* Generated images with given textual prompt (left two from DALL-E 3 (Betker et al., 2023), right two from FLUX (Labs, 2024)). Accurate orientation estimation is helpful to confirm whether generated contents follow the given orientation or perspective condition.

KITTI (Geiger et al., 2012), nuScenes (Caesar et al., 2020), Objectron (Ahmadyan et al., 2021) and ARKitScenes (Baruch et al., 2021), respectively. Our model can robustly and accurately predict the object orientation in images of various sources and resolutions.

## D. Visualization of Ori-Bench

All Ori-Bench samples, along with the responses from GPT-4o, Gemini-1.5-pro, and Orient Anything+LLM, are included in the attached file. We visualize the three kinds of subtasks in Ori-Bench in Fig. 13, 14 and 15, respectively.

Our observations reveal that these questions, which are intuitive for humans, often confuse the state-of-the-art VLM models like GPT-4o and Gemini-1.5-pro. This highlights the inherent limitations of existing approaches to understanding orientation. By utilizing the simple template to describe object orientations estimated by Orient Anything to LLM, we outperform alternative methods by a substantial margin.

## E. Orient Anything for Orientation Understanding

In Section 7.1 of the main text, we briefly introduce the use of Orient Anything for solving orientation understanding problems. Here, we provide a detailed implementation.

For the open domain orientation understanding problem, we first use LLM to extract the object nouns in the question, then use Grounding-SAM (Ren et al., 2024; Liu et al., 2025) to determine the coordinates of each object, and use Orient Anything to predict the horizontal orientation of each object. We convert the detected spatial information into text descriptions with simple templates. For multiple objects, we use their coordinates to express their left-right relationship in the image. For each object, we only consider the azimuth angle and convert it into the horizontal 8-direction description. Finally, we provide these templated spatial descriptions, questions, and options in LLM. Practical ex-

amples are provided in Fig. 13, 14, and 15.

Although this method has obvious disadvantages (ignoring depth and 3D object relationships), it still performs much better than the Gemini-1.5-pro and GPT-4o.

The template description of the object relationship is:

---

For **OBJ1** located in [x1, y1] with predicted azimuth angle $\hat{\varphi}$ and **OBJ2** located in [x2, y2],

*if x1 < x2:*
  "From the perspective of viewer <**OBJ1**> is on the left and <**OBJ2**> is on the right of the view."
  *if 292.5° < $\hat{\varphi}$ < 360° or 0° < $\hat{\varphi}$ < 67.5°:*
    "<**OBJ2**> is on the left of <**OBJ1**>."
  *if 67.5° < $\hat{\varphi}$ < 112.5°:*
    "<**OBJ2**> is behind <**OBJ1**>."
  *if 112.5° < $\hat{\varphi}$ < 247.5°:*
    "<**OBJ2**> is on the right of <**OBJ1**>."
  *if 247.5° < $\hat{\varphi}$ < 292.5°:*
    "<**OBJ2**> is in front of <**OBJ1**>."

*if x1 > x2:*
  "From the perspective of viewer <**OBJ2**> is on the left and <**OBJ1**> is on the right of the view."
  *if 292.5° < $\hat{\varphi}$ < 360° or 0° < $\hat{\varphi}$ < 67.5°:*
    "<**OBJ2**> is on the right of <**OBJ1**>."
  *if 67.5° < $\hat{\varphi}$ < 112.5°:*
    "<**OBJ2**> is in front of <**OBJ1**>."
  *if 112.5° < $\hat{\varphi}$ < 247.5°:*
    "<**OBJ2**> is on the left of <**OBJ1**>."
  *if 247.5° < $\hat{\varphi}$ < 292.5°:*
    "<**OBJ2**> is behind <**OBJ1**>."

---

The template description of object direction is:

For **OBJ** with predicted azimuth angle $\hat{\varphi}$
*if $292.5° < \hat{\varphi} < 360°$ or $0° < \hat{\varphi} < 22.5°$:*
  "The <**OBJ**> is facing the viewer."
*if $22.5° < \hat{\varphi} < 67.5°$:*
  "The <**OBJ**> is facing the viewer and to the left of the viewer."
*if $67.5° < \hat{\varphi} < 112.5°$:*
  "The <**OBJ**> is facing to the left of the viewer."
*if $112.5° < \hat{\varphi} < 157.5°$:*
  "The <**OBJ**> is facing away from the viewer and to the left of the viewer."
*if $157.5° < \hat{\varphi} < 202.5°$:*
  "The <**OBJ**> is facing away from the viewer."
*if $202.5° < \hat{\varphi} < 247.5°$:*
  "The <**OBJ**> is facing away from the viewer and to the right of the viewer."
*if $247.5° < \hat{\varphi} < 292.5°$:*
  "The <**OBJ**> is facing to the right of the viewer."
*if $292.5° < \hat{\varphi} < 337.5°$:*
  "The <**OBJ**> is facing the viewer and to the right of the viewer."

**Accurate Orientation Angles Estimation**: I will ask you a question about the content of the picture. Here is the question: <**image**> Align the front of the object towards the viewer. Rotate the object x degrees to its right (i.e., clockwise from a top view), using a 360° per full circle unit system. Adjust the height of the viewer to form a pitch Angle y with the object (same unit of degrees; y is positive if the viewer is looking down at the object, and y is negative if the viewer is looking up at the object). Finally, the viewer is rotated clockwise by an Angle z (same unit of degrees) with the line connecting the viewer and the object as the axis, and a negative z indicates a counterclockwise rotation. Now, please directly predict the values of x, y, and z in float format.

## F. Prompts for VLMs

**Question Answering for Ori-Bench and Orientation Recognition**: I will ask you a single-choice question about the content of the picture. Here is the question: <**image**> <**question**> <**options**>.

**Orientation Annotating for Orthogonal Rendering views**: I'm going to show four images of the same object from four viewpoints in turn and label them 'A.' 'B.' 'C.' 'D.' Four options. Option 'E.' is "No front face or More than One front Face". Decide whether it has a front and if yes, which one is the front of the object after the presentation. Note that: If the object is a gun, bow and arrow, etc., please use the muzzle of the gun as the front. Stick tools and weapons such as swords, axes, knives, and wrenches are considered to have no front. If you cannot decide or there is more than one front, you should choose 'E.'. A.<**image viewA**> B.<**image viewB**> C.<**image viewC**> D.<**image viewD**> E.No front face.

| Category | Cube RCNN | Gemini | GPT-4o | Orient Anything (ViT-L) |
|---|---|---|---|---|
| bed | 75% | 15% | 40% | 100%(+25%) |
| monitor | 35% | 50% | 50% | 100%(+50%) |
| oven | 50% | 10% | 65% | 100%(+35%) |
| teddy bear | 20% | 40% | 45% | 100%(+55%) |
| motorbike | 5% | 20% | 40% | 95%(+55%) |
| parking meter | 40% | 55% | 65% | 95%(+30%) |
| laptop | 65% | 45% | 50% | 95%(+30%) |
| sheep | 15% | 45% | 45% | 90%(+45%) |
| elephant | 5% | 30% | 55% | 90%(+35%) |
| sofa | 5% | 25% | 50% | 90%(+40%) |
| toilet | 55% | 20% | 50% | 90%(+35%) |
| cell phone | 35% | 75% | 80% | 90%(+10%) |
| microwave | 35% | 25% | 50% | 90%(+40%) |
| clock | 20% | 45% | 60% | 90%(+30%) |
| bus | 10% | 20% | 40% | 85%(+45%) |
| traffic light | 0% | 35% | 50% | 85%(+35%) |
| stop sign | 0% | 70% | 75% | 85%(+10%) |
| bench | 20% | 20% | 20% | 85%(+65%) |
| bear | 5% | 30% | 40% | 85%(+45%) |
| zebra | 5% | 30% | 50% | 85%(+35%) |
| sink | 0% | 0% | 30% | 85%(+55%) |
| cat | 20% | 45% | 60% | 80%(+20%) |
| dog | 10% | 35% | 60% | 80%(+20%) |
| horse | 10% | 50% | 35% | 80%(+30%) |
| chair | 10% | 20% | 30% | 80%(+50%) |
| book | 45% | 80% | 45% | 80%(+0%) |
| car | 10% | 40% | 45% | 75%(+30%) |
| truck | 15% | 35% | 60% | 75%(+15%) |
| cow | 20% | 40% | 40% | 75%(+35%) |
| person | 5% | 35% | 40% | 70%(+30%) |
| aeroplane | 15% | 20% | 60% | 70%(+10%) |
| refrigerator | 20% | 25% | 55% | 70%(+15%) |
| bird | 10% | 25% | 60% | 65%(+5%) |
| giraffe | 15% | 30% | 55% | 65%(+10%) |
| train | 30% | 15% | 60% | 60%(+0%) |
| fire hydrant | 20% | 20% | 30% | 55%(+25%) |
| boat | 10% | 25% | 45% | 50%(+5%) |
| backpack | 40% | 65% | 50% | 50%(-15%) |
| mouse | 15% | 0% | 0% | 50%(+35%) |
| kite | 5% | 40% | 60% | 45%(-15%) |
| hair drier | 0% | 20% | 55% | 45%(-10%) |
| bicycle | 5% | 30% | 30% | 40%(+10%) |
| toaster | 50% | 25% | 30% | 40%(-10%) |
| remote | 10% | 5% | 10% | 5%(-5%) |
| keyboard | 10% | 0% | 0% | 0%(-10%) |

*Table 7.* Detailed horizontal direction recognition accuracy for each object category in COCO that is annotated with front face and orientation. The differences between Orient Anything and the best results achieved by other alternative methods are also provided.

Cube RCNN · Orient Anything

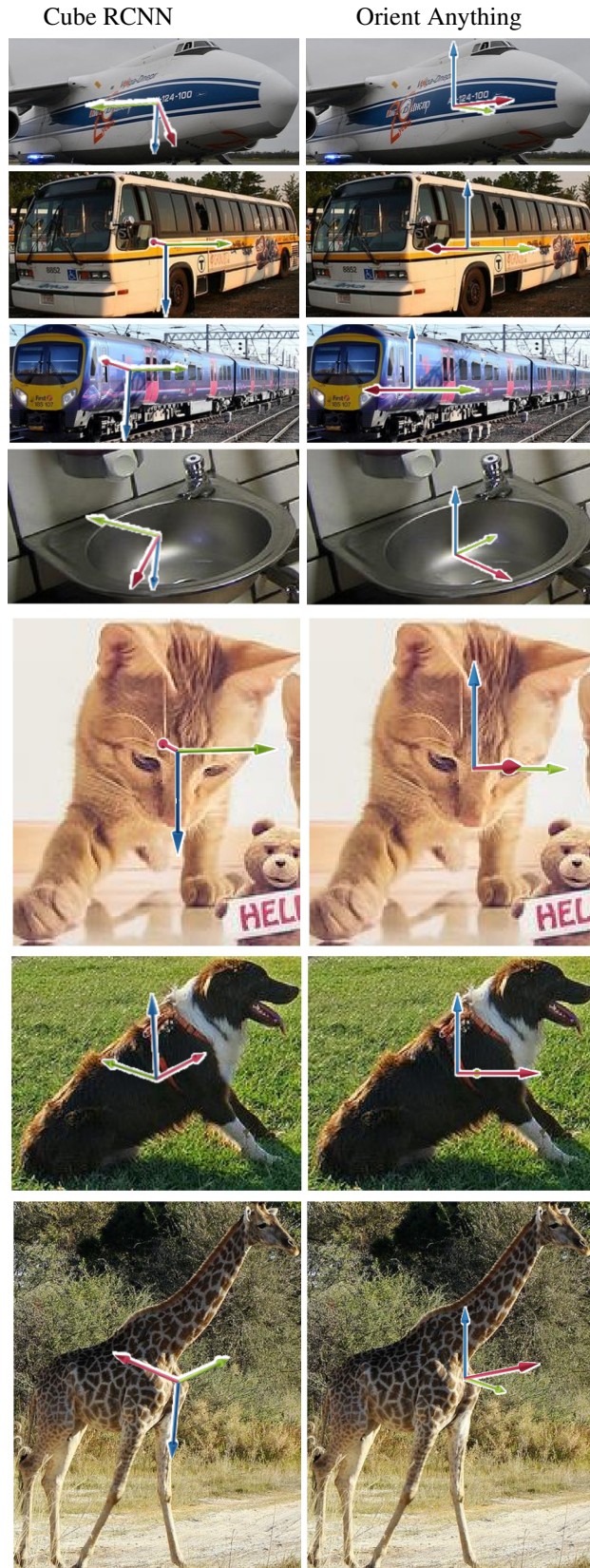

*Figure 8.* Qualitative results on COCO

Ground Truth     Cube RCNN     Orient Anything

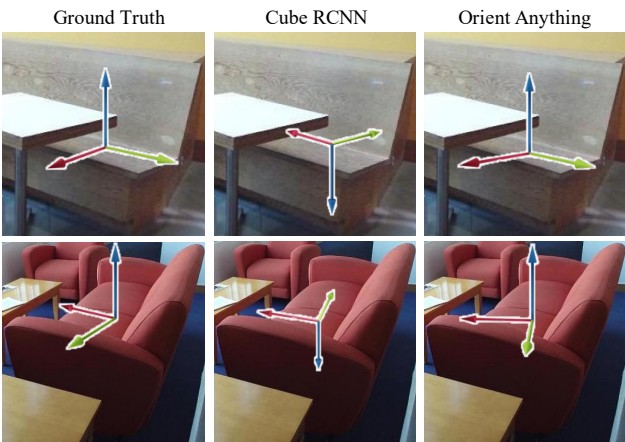

*Figure 9.* Qualitative results on SUN RGB-D.

Ground Truth     Cube RCNN     Orient Anything

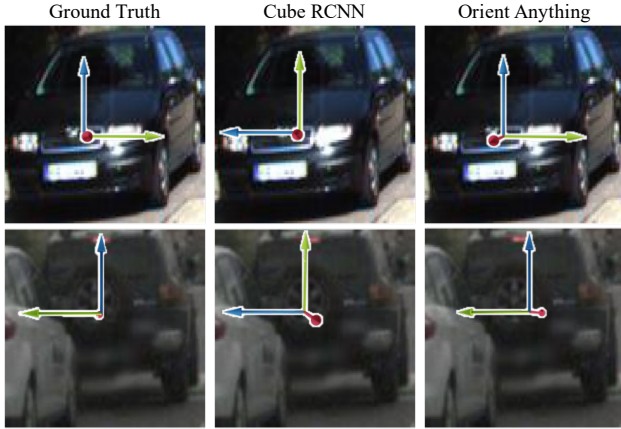

*Figure 10.* Qualitative results on KITTI and nuScenes.

Ground Truth     Cube RCNN     Orient Anything

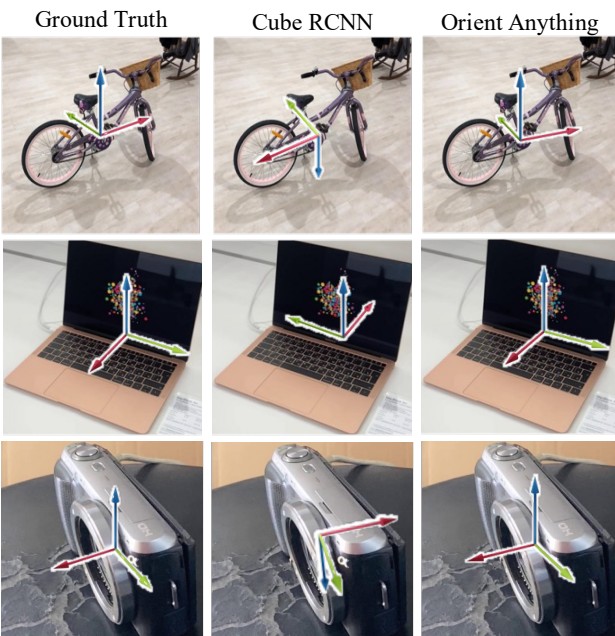

*Figure 11.* Qualitative results on Objectron.

Ground Truth     Cube RCNN     Orient Anything

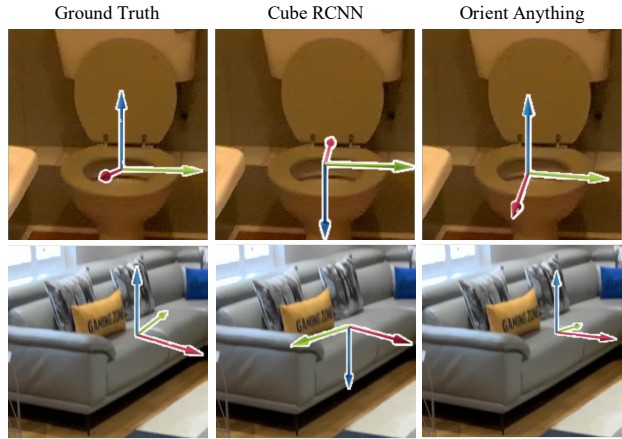

*Figure 12.* Qualitative results on ARKitScenes.

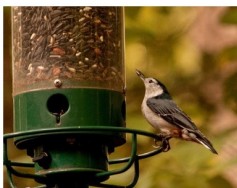

Which side is the animal facing in the picture?
A. Facing the viewer
B. Facing away from the viewer
C. Facing left
D. Facing right

**GPT-4o**: D. The bird is oriented so that its tail is away from the viewer and its head is pointed to the right of the frame.

**Gemini-1.5-pro**: C. The bird's beak and head are pointing towards the left side of the image.

Spatial Context: 1. animal is facing to the left of the viewer.
**Orient Anything+LLM**: C. The animal is described as facing to the left of the viewer. This implies its left side is towards the viewer's left.

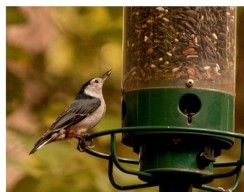

Which side is the animal facing in the picture?
A. Facing the viewer
B. Facing away from the viewer
C. Facing left
D. Facing right

**GPT-4o**: C. The bird's head is turned towards the left side of the image.

**Gemini-1.5-pro**: D. The bird's beak and head are pointing to the right side of the image. Its tail feathers point to the left.

Spatial Context: 1. animal is facing to the right of the viewer.
**Orient Anything+LLM**: D. The animal is described as facing to the right of the viewer, meaning its front is towards the right.

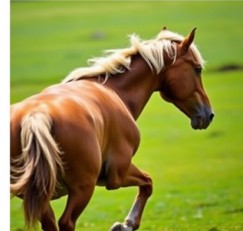

Which orientation the horse is facing to?
A. Facing the viewer while slightly to the right
B. Facing the viewer while slightly to the left
C. Facing away from the viewer while slightly to the right
D. Facing away from the viewer while slightly to the left

**GPT-4o**: D. The horse's head is turned to the left and the body faces the direction away from the viewer.

**Gemini-1.5-pro**: D. The horse's rear is towards the viewer and its head is slightly turned to the left.

Spatial Context: 1. horse is facing away from the viewer and to the right of the viewer.
**Orient Anything+LLM**: C. The question states that the horse was facing away from the viewer and to the right of the viewer. This directly matches the description of option C.

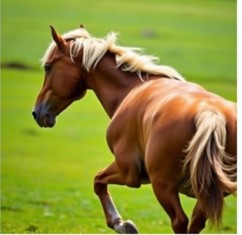

Which orientation the horse is facing to?
A. Facing the viewer while slightly to the right
B. Facing the viewer while slightly to the left
C. Facing away from the viewer while slightly to the right
D. Facing away from the viewer while slightly to the left

**GPT-4o**: D. The horse's head is pointed to the left of the image.

**Gemini-1.5-pro**: D. The horse's head is turned towards the left side of the image, which indicates that its body is oriented slightly to the left while facing away from the viewer.

Spatial Context: 1. horse is facing away from the viewer and to the left of the viewer.
**Orient Anything+LLM**: D. The horse is described as facing away from the viewer and to the left of the viewer. This aligns with option D.

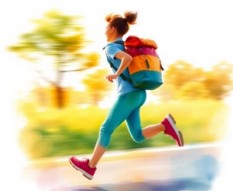

Which side is the girl facing in the picture?
A. Facing the camera
B. Facing away from the camera
C. Facing left
D. Facing right

**GPT-4o**: D. The girl is running and her body is angled towards one side.

**Gemini-1.5-pro**: C. The girl's face is visible, but her body is oriented to the left of the frame, indicating that she is running towards the left side of the image.

Spatial Context: 1. girl is facing to the left of the viewer.
**Orient Anything+LLM**: C. Since the girl is facing to the left of the viewer, she is turned to her left from the perspective of the camera/viewer.

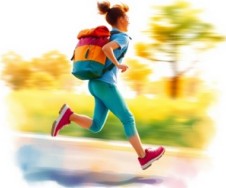

Which side is the girl facing in the picture?
A. Facing the camera
B. Facing away from the camera
C. Facing left
D. Facing right

**GPT-4o**: D. The girl appears to be moving towards the right side of the image.

**Gemini-1.5-pro**: D. The girl's face is visible, and her body is oriented towards the right side of the image.

Spatial Context: 1. girl is facing to the right of the viewer.
**Orient Anything+LLM**: D. The girl is facing to the right of the viewer, meaning from the viewer's perspective, she is turned to her own left side, which corresponds to option D.

*Figure 13.* Visualization and qualitative comparison on the *Object Direction Recognition* task of Ori-Bench.

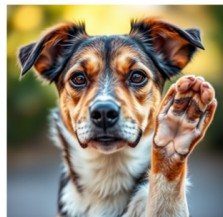

Which paw is the dog raising?
A.its left front paw
B.its right front paw

**GPT-4o**: B. The visible paw appears to be on the right side from the dog's perspective, indicating it's the right front paw.

**Gemini-1.5-pro**: A. The dog's right paw would be on the left side of the image from the viewer's perspective. The dog is raising the paw that appears on the right side of the image, which is its left paw.

Spatial Context: 1. dog is facing the viewer. 2. From the perspective of viewer, dog is on the left and paw is on the right of the view. 3. From the perspective of dog, paw is on the left of dog.
**Orient Anything+LLM**: A. From the viewer's perspective, the dog is facing them, with its paw also facing them. This implies the viewer is looking directly at the dog. The viewer perceives the paw on the right. From the dog's perspective, this would be its left paw.

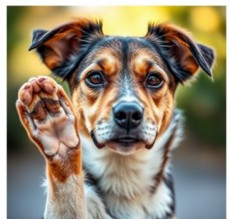

Which paw is the dog raising?
A.its left front paw
B.its right front paw

**GPT-4o**: A. The paw shown is on the left side of the image.

**Gemini-1.5-pro**: A. The dog's head is tilted slightly to the right, indicating that the paw being raised is on its left side. Also, considering the camera's perspective, the paw visually closer to the viewer would be the left paw.

Spatial Context: 1. dog is facing the viewer. 2. From the perspective of viewer, paw is on the left and dog is on the right of the view. 3. From the perspective of dog, paw is on the right of dog.
**Orient Anything+LLM**: B. Paw is facing the viewer and is on the left from the viewer's perspective, but on the right from the dog's perspective, meaning it is the dog's right front paw.

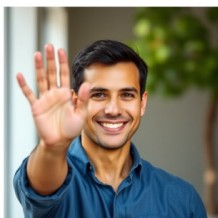

Which hand is the man raising?
A.his left hand
B.his right hand

**GPT-4o**: A. The thumb is on the right side of the raised hand, indicating it is his left hand.

**Gemini-1.5-pro**: B. The man's right thumb is on the left side of the image, indicating that it is his right hand raised.

Spatial Context: 1. man is facing the viewer. 2. From the perspective of viewer, hand is on the left and man is on the right of the view. 3. From the perspective of man, hand is on the left of man.
**Orient Anything+LLM**: B. From the man's perspective, the hand that is on the right is his right hand. Since the viewer sees the hand on the left, the man would be raising his right hand.

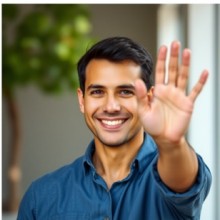

Which hand is the man raising?
A.his left hand
B.his right hand

**GPT-4o**: B. The man is raising the hand that appears on the right side of the image, which corresponds to his right hand.

**Gemini-1.5-pro**: B. The man is raising the hand that is on the right side of the image, from the viewer's perspective. This is his right hand.

Spatial Context: 1. man is facing the viewer. 2. From the perspective of viewer, man is on the left and hand is on the right of the view. 3. From the perspective of man, hand is on the left of man.
**Orient Anything+LLM**: A. From the perspective of the man, the hand on his left is being mentioned. Since the man is facing the viewer and his left hand is on the viewer's right, it means he is raising his left hand.

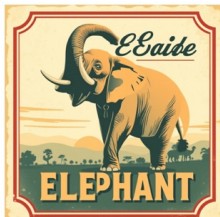

Which leg of the elephant is closest to the camera?
A.its left front
B.its right front
C.its left rear
D.its right rear

**GPT-4o**: A. The elephant's left front leg appears prominently in the foreground.

**Gemini-1.5-pro**: B. The elephant's right front leg is closest to the camera. We can tell because the perspective of the image makes it appear larger and more prominent than the left front leg.

Spatial Context: 1. elephant is facing the viewer and to the left of the viewer.
**Orient Anything+LLM**: A. The elephant is facing the viewer, and its left side is also closest to the viewer. Therefore, the leg closest to the camera is its left front leg. A

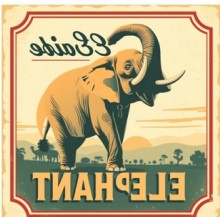

Which leg of the elephant is closest to the camera?
A.its left front
B.its right front
C.its left rear
D.its right rear

**GPT-4o**: A. The illustration shows the left front leg as most prominent and forward.

**Gemini-1.5-pro**: B. The elephant's left front leg is partially obscured by its trunk and head, making the right front leg clearly the closest to the camera.

Spatial Context: 1. elephant is facing the viewer and to the right of the viewer.
**Orient Anything+LLM**: B. Since the elephant is facing the viewer and to the right of the viewer, its right side is closer to the camera. Among the given options, its right front leg is the closest.

*Figure 14.* Visualization and qualitative comparison on the *Spatial Part Reasoning* task of Ori-Bench.

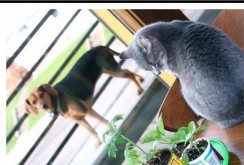

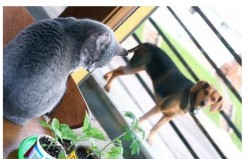

From the dog's perspective, is the cat on its left or right?
A.Left
B.Right

**GPT-4o**: A. The cat is positioned closer to the window and is facing the dog, putting the cat on the dog's left side.

**Gemini-1.5-pro**: B. The dog is looking at the cat and the cat is looking down. To the dog's perspective, the cat is on its right.

Spatial Context: 1. dog is facing the viewer and to the left of the viewer. 2. cat is facing away from the viewer and to the left of the viewer. 3. From the perspective of viewer, dog is on the left and cat is on the right of the view. 4. From the perspective of dog, cat is on the left of dog.
**Orient Anything+LLM**: A. Since the dog is facing the viewer, to find the dog's perspective, one can imagine turning around such that the dog is looking at the viewer. From this angle, the cat, which is to the viewer's right, would then logically be on the dog's left.

From the dog's perspective, is the cat on its left or right?
A.Left
B.Right

**GPT-4o**: B. The cat is positioned on the right side of the image as viewed from the dog's perspective.

**Gemini-1.5-pro**: B. The dog is facing the sliding door. The cat is on the other side of the glass. From the dog's perspective, the cat is to its right.

Spatial Context: 1. dog is facing the viewer. 2. cat is facing away from the viewer and to the right of the viewer. 3. From the perspective of viewer, cat is on the left and dog is on the right of the view. 4. From the perspective of dog cat is on the right of dog.
**Orient Anything+LLM**: B. The dog is facing the viewer, so from the dog's perspective, the left and right are inverted compared to the viewer's perspective. Therefore, the cat, which is to the left from the viewer's perspective, is on the right from the dog's perspective.

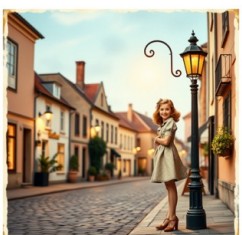

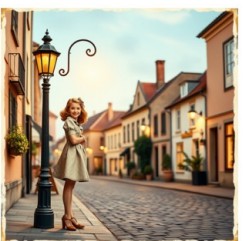

Will the girl run into the lamp post if she walks forward?
A.Yes
B. No

**GPT-4o**: A. The girl is standing close to the lamp post and facing it, so if she walks directly forward, she is likely to run into it. A

**Gemini-1.5-pro**: B. The girl is standing next to the lamp post, facing the camera. If she walks forward, she will walk away from the lamp post, down the street.

Spatial Context: 1. girl is facing to the left of the viewer. 2. From the perspective of viewer, girl is on the left and lamp post is on the right of the view. 3. From the perspective of girl, lamp post is behind girl.
**Orient Anything+LLM**: B. From the perspective of the girl, the lamp post is behind her. So, if she walks forward, she will not run into it.

Will the girl run into the lamp post if she walks forward?
A.Yes
B. No

**GPT-4o**: B. The girl is standing beside the lamp post, not directly in front of it. B

**Gemini-1.5-pro**: B. The girl is standing next to the lamp post, facing the street. If she walks forward, she will be walking away from the lamp post, down the street.

Spatial Context: 1. girl is facing to the right of the viewer. 2. From the perspective of viewer lamp post is on the left and girl is on the right of the view. 3. From the perspective of girl, lamp post is behind girl.
**Orient Anything+LLM**: B. Since lamp post is behind the girl from her perspective, walking forward means moving away from the lamp post.

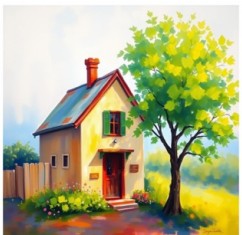

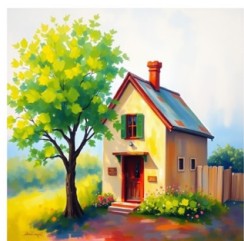

If someone walks out the door, on which side would the tree be relative to them?
A.left side
B.right side

**GPT-4o**: A. The tree is to the left of the house entrance.

**Gemini-1.5-pro**: B. The tree is located on the right side of the house, as observed from the front door.

Spatial Context: 1. house is facing the viewer and to the right of the viewer. 2. From the perspective of viewer, house is on the left and tree is on the right of the view. 3. From the perspective of house, tree is on the left of house.
**Orient Anything+LLM**: A. The person walking out of the house will have the perspective of the house. Since the tree is on the left of the house from this perspective, the tree will be on the left side.

If someone walks out the door, on which side would the tree be relative to them?
A.left side
B.right side

**GPT-4o**: A. The tree is on the left side when facing the house from the outside, so as someone exits, it would be on their left side.

**Gemini-1.5-pro**: A. The tree is on the left side of the door from the perspective of someone exiting the house.

Spatial Context: 1. house is facing the viewer. 2. From the perspective of viewer, tree is on the left and house is on the right of the view. 3. From the perspective of house, tree is on the right of house.
**Orient Anything+LLM**: B. If the house is facing the viewer, and the tree is on the right from the house's perspective, then someone walking out the door will have the tree on their right side. B

*Figure 15.* Visualization and qualitative comparison on the *Spatial Relation Reasoning* task of Ori-Bench.

