# OpenReview forum: "Orient Anything: Learning Robust Object Orientation Estimation from Rendering 3D Models"
_ICML.cc/2025/Conference — ICML 2025 poster_

### Official Review · Reviewer_83jQ · 2025-03-10

**Overall Recommendation:** 3

**Summary:**

This paper proposes orient anything, a foundation model for predicting object orientation from a single image in an zero-shot manner. The key of the paper is curating a large-scale dataset for the orientation estimation task, which is rendered from Objaverse including 2M images. The authors propose to deal with the ambiguity of low-dimensional poses by using probabilistic distribution over object azimuth, elevation/polar angle, and camera rotation. The proposed method is evaluated on both synthetic images (on orientation error) and real images (on recognition tasks).

**Claims And Evidence:**

The authors claimed this paper to be the first foundation model for object orientation estimation. While it seems interesting, I am worried about the evaluation protocol on real data, where the orientation error is not studied.

**Essential References Not Discussed:**

N/A

**Experimental Designs Or Analyses:**

I am not fully convinced with the evaluations, especially the lack of results on orientation error on real images. Showing the recognition task performance is not enough for convincing me that this is an foundation model. And I am not very sure how the method trained on synthetic data with white background can generalize to real and even in-the-wild images.

Besides, one of the biggest contribution of this paper is the curated training data. However, there is no study on the dataset quality, even no visualization on how the dataset look like. Besides, I think it would be better if the authors perform an experiment on using different ratio of the curated dataset for training and explore whether there can be any scaling effect.

**Methods And Evaluation Criteria:**

The proposed methods include two parts: 1) data curation and 2) training an orientation prediction model. The data curation method includes filtering non-axis-aligned objects, identifying ambiguous symmetry objects, and rendering synthetic data. The orientation prediction model is based on a ViT encoder with several decoders for predicting the probabilistic distribution of rotation angles.

The trained model is evaluated on both real and synthetic data. On synthetic data, it is evaluated mainly with orientation error; while on real data, due to the lack of pose annotations, it is evaluated with higher-level recognition tasks.

**Other Comments Or Suggestions:**

N/A

**Other Strengths And Weaknesses:**

Please see comments above.

**Questions For Authors:**

1. Why not including quantitative results on real images? The authors claim that there are no data for performing evaluation. However, I don't think this is true. There are bunch of object pose estimation benchmarks that can be used for evaluation. Besides, the object 6D pose estimation works should also be discussed in details -- how the orientation estimation task is different, and why using no reference mesh/image is important.

2. Object orientation is not a well-defined task. The definition of orientation is closely related to the semantic meaning of the specifc object categories, i.e., the canonical pose space, where the objects are aligned as forward-facing. Thus, on novel categories, the orientation of objects are not well-defined. Thus, it is necessary to study how the model generalize to categories that are unseen during training, as the categories in objaverse is not guaranteed to include every category in real life.

3. The authors should also discuss why the model trained on synthetic data can generalize to real data.

3. Besides, the authors should study the quality of the curated dataset, and study the influence of training data scale.

**Relation To Broader Scientific Literature:**

The posed task can be useful for object 6D pose estimation and may have potential to benefit robotic applications.

**Theoretical Claims:**

N/A

---

> ### Author Rebuttal · Authors · 2025-04-01
>
> Thank you for your valuable review and suggestions. Below we respond to the comments in **Weaknesses (W)** and **Questions (Q)**.
>
> ---
>
> ***W1.1 & Q1.1: Quantitative orientation error on real images.***
>
> Currently, the 6 benchmarks in Table 2 are evaluated on real-world data with quantitative 3D orientation error.
>
>
>
> Moreover, we have also tested on the latest and largest object 3D orientation benchmark, ImageNet3D [1], which covers 200 object categories. The Acc@30° results are as follows:
> |Setting | Model                             | Avg. | Electronics | Furniture | Household | Music | Sports | Vehicles | Work |
> |---| --------------------------------- | ---- | ----------- | --------- | --------- | ----- | ------ | -------- | ---- |
> |Zero-shot| ImageNet3D-ResNet50   | 37.1 | 30.1        | 35.6      | 28.1      | 11.8  | **51.7** | 36.7     | **40.9** |
> || Orient Anything-B     | **48.5** | **61.0**    | **66.8**  | **37.9**  | **27.3** | 25.6     | **70.8** | 33.4     |
> |Fine-tuning| ImageNet3D-ResNet50 | 53.6 | 49.2        | 52.4      | 45.8      | 26.0  | 65.2   | 56.5     | 58.5 |
> || ImageNet3D-DINOv2-B | 64.0 | 75.3        | 47.9      | 32.9      | 23.5  | **74.7** | 38.1     | **64** |
> || Orient Anything-B     | **71.3** | **77.6**    | **89.7**  | **64.4**  | **54.4** | 47.6   | **87.4** | 61.2 |
>
> Note that we couldn't find detailed definitions for the major categories like *Electronics*, *Furniture*, and *Household* in ImageNet3D, so we used GPT to map its 200 categories into the 7 general categories. As a result, we used GPT to map its 200 categories into 7 broader ones. Therefore, the comparison results for each general category may vary, and the average score provides a more meaningful comparison.
>
> [1] Ma W, Zhang G, Liu Q, et al. Imagenet3d: Towards general-purpose object-level 3d understanding. NeurIPS 2024.
>
> ---
>
> ***W1.2: Generalization to real and even in-the-wild images.***
>
> We provide numerous visualizations on real and in-the-wild images in both our main text and supplementary materials. Furthermore, in response to your W1.1 & Q1.1, we present the results on the current largest 3D orientation estimation benchmark, ImageNet3D, which also demonstrates strong generalization to these real-world scenarios.
>
> ---
>
> **W2.1 & Q4.1: Study on the dataset quality.**
>
> In <https://anonymous.4open.science/r/visualization-B728/Training_Samples/>, we provide lots of visualization cases from our curated training dataset, showcasing both high-quality images and accurate orientation annotations.
>
> ---
>
> ***W2.2 & Q4.2: Influence of training data.***
>
> We train ViT-B version models using different ratios of data. Below are the results:
>
> | Data Ratio | COCO(Acc) | ImageNet3D(Acc@30°) |
> | ---------- | --------- | ------------------- |
> | 25%        | 63.55     | 45.47               |
> | 50%        | 65.47     | 44.08               |
> | 75%        | 66.82     | 47.65               |
> | 100%       | **69.85** | **48.52**           |
>
>  ---
>
> ***Q1.2 Difference and advantage to object 6D pose estimation.***
>
> **Difference:** Traditional 6D pose estimation methods focus on relative orientation to a reference frame or template 3D model, while Orient Anything focuses on semantic orientation (e.g., the semantic “front face” of an object) without any reference. Therefore, the previous benchmark for object pose estimation is not suitable for our task.
>
>
>
> **Importance of no reference mesh/image:** Monocular images are the most accessible and widely used form of visual input, and in many scenarios, references for the desired object are often unavailable. By not relying on reference meshes or images, Orient Anything enables broader applications, such as solving spatial reasoning questions and evaluating whether the generated image adheres to the desired spatial relationships, as discussed in Section 7.
>
>
>
> ---
>
> ***Q2 Generalization to unseen categories.***
>
> In response to your W1.1 & Q1.1, we further provide evaluation results on the current largest single-view orientation estimation benchmark, ImageNet3D, which significantly surpasses existing methods and demonstrates strong generalization to real images and various categories.
>
> Additionally, in response to Reviewer ev5K’s W1 & Q1, we discussed how to further scale the annotated data and expand the covered categories through synthetic 3D assets and voting-based annotation strategy.
>
> ---
>
> ***Q3 Why trained on synthetic data can generalized to real data.***
>
> The synthetic-to-real generalization ability is mainly obtained through the task-agnostic pre-trained model that is trained on massive real images. The similar idea has been discussed and validated in Marigold [2] and Depth Anything V2 [3]. For further discussions, please refer to the response to Reviewer dtES’s W3.
>
> [2] Ke B, Obukhov A, Huang S, et al. Repurposing diffusion-based image generators for monocular depth estimation. CVPR 2024
>
> [3] Yang L, Kang B, Huang Z, et al. Depth anything v2. NIPS 2024

---

> > ### Comment · Reviewer_83jQ · 2025-04-04
> >
> > Thanks for the detailed rebuttal and extra evaluation results. The rebuttal have addressed all of my questions, and I have updated my score.

---

> > > ### Author Response · Authors · 2025-04-05
> > >
> > > We sincerely appreciate your kind support! In the final revision, we will further enhance the paper by incorporating the additional experimental results and the valuable insights from the reviews. Thank you again!

---

### Official Review · Reviewer_dtES · 2025-03-13

**Overall Recommendation:** 3

**Summary:**

The paper introduces Orient Anything, a foundational model designed for zero-shot estimation of object orientation from monocular images. Due to the scarcity of orientation annotations for open-world objects, the authors develop an automated 3D object orientation annotation pipeline that effectively utilizes the extensive resources of 3D models to create a diverse dataset of 2 million images with precise orientation labels. To enhance training stability and improve generalization, the authors introduce a robust training objective that models the 3D orientation as a probability distribution. Additionally, they propose several strategies to improve synthetic-to-real transfer, achieving state-of-the-art accuracy in orientation estimation for both rendered and real images. Experimental results demonstrate the superiority of the proposed method and highlight the significant potential of Orient Anything for high-level applications, such as enhancing spatial understanding and scoring spatial generation.

**Claims And Evidence:**

The claims made in the submission are well supported by clear and convincing evidence.

**Essential References Not Discussed:**

Essential References have been well discussed in the paper.

**Experimental Designs Or Analyses:**

The experimental design and analysis are generally sound.
Table 2&3 show the superiority of the proposed method on both in-domain and out-of-domain datasets.
Table 4&5 fully verify the key designs of the proposed method.

Besides, I still have some questions:
1. In figures 13 and 14, it appears that the input spatial context provided for 'Orient Anything+LLM' already contains the answer to the question. This raises a question about whether it is valid to assess the LLM's response in this scenario. Does the improvement of 'Orient Anything+LLM' shown in Table 1 primarily rely on the accuracy of Orient Anything itself?
2. The example shown in figure2 is inappropriate. The author does not provide a clear coordinate reference for LLM, which makes the answer ambiguous. After trying the following two questions, GPT-4o got the correct answer:
Q1: Does Falcon face to me? & Q2: So, in Falcon's view, where is Captain America.

**Methods And Evaluation Criteria:**

The proposed method holds significant value for a variety of applications, including understanding spatial information from images, verifying the spatial accuracy of generated images, and more.

My main concern is the weak novelty of the method itself. Most of the designs are common in computer vision.

Nevertheless, I think the contribution of this article in terms of dataset and foundation model is still meaningful.

**Other Comments Or Suggestions:**

Can the method proposed in this paper be applied to video data? Additionally, is it capable of achieving stable orientation estimation results from video content?

**Other Strengths And Weaknesses:**

Strengths:
1. The ablation experiments in this paper are very sufficient. They fully verify the main design and contribution of this paper.
2. This paper is well-written and easy to follow.
3. The created large dataset is of great significance to the development of the community.

Main weaknesses:
1. The experiments and analysis on LLMs are biased as discussed in 'Experimental Designs Or Analyses'. The authors should provide a more comprehensive comparison and analysis. For example, increasing the complexity of the question to weaken its connection with the given spatial context.
2. The use of 'Synthetic-to-Real Transferring' in the paper lacks rigor. Firstly, employing a better pre-trained model can enhance the model's performance on its own, which is not directly related to synthetic-to-real transfer. Secondly, the crop operation is a standard procedure in deep learning.
3. What is the speed and cost of the method?
4. Compared with the 6D pose estimation methods (e.g., FoundationPose), does the proposed method have more advantages in orientation estimation?

**Questions For Authors:**

No more questions.

**Relation To Broader Scientific Literature:**

The model proposed in this paper can evaluate the spatial correctness of the results of the generative model. As discussed in "Towards Foundation Models for 3D Vision: How Close Are We?", state-of-the-art Vision-Language Models still perform poorly for spatial understanding, which is consistent with the findings of this paper.

**Theoretical Claims:**

I have check the correctness of equations(1-4) in section 5.1 and have no issues about them.

---

> ### Author Rebuttal · Authors · 2025-04-01
>
> Thank you for your positive review for recognizing the significance of our paper and invaluable suggestions. Below we respond to the comments in **Weaknesses (W)** and **Questions (Q)**.
>
> ---
>
> ***Q1: Improvement of "Orient Anything+LLM".***
>
> "Orient Anything + LLM" is designed to demonstrate the accuracy of Orient Anything in real-world applications and highlight the importance of orientation perception for understanding spatial relationships. Since we use a text-based LLM that cannot perceive visual content, it relies on Orient Anything and the object detection model to understand spatial relationships in images. The improvements indeed come from the accuracy of these models.
>
> ---
>
> ***Q2: Example shown in Figure2.***
>
> First, we conducted the test in November 2024, and subsequent version updates of GPT-4o may result in different results.
>
> Second, we believe the phrase "In Falcon's view (or perspective)" provides a coordinate reference, making both our question and your Q2 unambiguous. Adding a prefix question (e.g., "Does Falcon face me?") may explicitly prompt the model to transform the perspective first, which should be a natural reasoning step in a single Q&A. Breaking down a question into sub-questions natively lowers its difficulty, so comparing direct answers with manually split versions may not be entirely fair.
>
> ---
>
> ***W1: Increasing the complexity of the question.***
>
> When constructing Ori-Bench, we created different subsets for distinct purposes. The basic "object direction" subset directly tasks the model with recognizing the orientation of individual objects, which aligns with the capabilities of Orient Anything. The more advanced "spatial part" subset requires understanding the pose of individual objects. Finally, the "spatial relation" subset involves open-domain questions that require further reasoning to solve.
>
>
>
> We believe these three progressive subsets offer a more comprehensive evaluation of a model's ability to address orientation-related questions. The most challenging "spatial relation" subset contains the most test samples, and improvements in this subset highlight the importance of understanding orientation when handling high-level spatial reasoning tasks.
>
>
>
> Thank you very much for your suggestion. In future updates, we will include more challenging questions and try to distill Orient Anything's knowledge into a VLM, thereby inherently integrating object orientation understanding capabilities.
>
> ---
>
> ***W2: Discussion about “Synthetic-to-Real Transferring”.***
>
> Several works have discussed the relationship between pre-trained models and synthetic-to-real transfer. Marigold[1] transforms Stable Diffusion into a depth diffusion model using only synthetic data. Depth Anything v2[2] provides a more systematic analysis of how different pre-trained models impact "Synthetic-to-Real Transfer."
>
>
>
> Additionally, while the crop operation is a standard processing technique, in our case, it is specifically aligned with the goal of transferring from (complete) synthetic training objects to (often occluded) real objects.
>
> [1] Ke B, Obukhov A, Huang S, et al. Repurposing diffusion-based image generators for monocular depth estimation. CVPR 2024
>
> [2] Yang L, Kang B, Huang Z, et al. Depth anything v2. NIPS 2024
>
>
>
> ---
>
> ***W3: Speed and cost of Orient Anything.***
>
> We only add four very lightweight MLP prediction heads (about 1M parameter) to the standard DINOv2 encoder. The parameter number remains nearly identical to the original DINOv2.
>
> |              | DINOv2 Encoder | Project Heads |
> | ------------ | -------- | ------------ |
> | Orient-Anything-S | 22.06M   | 1.17M        |
> | Orient-Anything2-ViT-B | 86.58M   | 1.51M        |
> | Orient-Anything-ViT-L | 304.37M  | 1.74M        |
>
> ---
>
> ***W4: Difference and Advantage over 6D pose estimation method (e.g., FoundationPose).***
>
> **Difference:** Traditional 6D pose estimation methods focus on relative orientation to a reference frame or template 3D model, while Orient Anything focuses on semantic orientation (e.g., the semantic “front face” of an object) without any reference.
>
> **Advantage:** Orient Anything does not require multi-view reference images or a ground-truth 3D model during inference. The predicted orientation is inherently linked to the object's semantic front, enabling broader applications, such as enhancing spatial understanding in VLMs and image generation.
>
>  ---
>
> **Other: Extend to video.**
>
> Please refer to the response to Reviewer MxHL’s Q4.

---

> > ### Comment · Reviewer_dtES · 2025-04-05
> >
> > Thanks for the rebuttal of authors which addressed most of my questions. Here, I still have some concerns:
> > 1. The so-called contribution of 'Sythetic-to-Real Transferring' is over-claimed! As discussed by the authors, this idea has already been verified in previous works.
> > 2. As shown in 'W2.2 & Q4.2: Influence of training data' of Reviewer 83jQ, the performance on ImageNet3D improves slightly as the amount of training data increases. This makes me worry about the room for improvement of the model's zero-shot performance.

---

> > > ### Author Response · Authors · 2025-04-05
> > >
> > > Thanks for your response and further comments!
> > >
> > >  ---
> > >
> > > ***Concern 1: Over-claimed 'Sythetic-to-Real Transferring'.***
> > >
> > > Our strategy for “Synthetic-to-Real Transferring” involves two key components:
> > > - Model initialization to inherit real-world knowledge, and
> > > - Data augmentation to narrow the domain gap.
> > >
> > > Regarding *model initialization*, we fully acknowledge that this idea has been explored in prior work, as stated in the second paragraph of Section 5.2. Besides, we explicitly clarify that we are *“evaluating this idea in our orientation estimation tasks”* in line 298. We sincerely appreciate your thoughtful reminder and will revise the description of this aspect in the final revision to ensure our claims are well-calibrated and clearly clarified.
> > >
> > > As for *data augmentation*, our crop-based training and segment-based testing augmentations are carefully designed to match the specific distributional differences between rendered and real-world images. These augmentations are tightly integrated with our methods and data.
> > >
> > > ---
> > >
> > > ***Concern 2: Slight improvement on ImageNet3D with data scaling.***
> > >
> > > This observation is largely due to **differences in how object orientation is defined** in our dataset versus ImageNet3D for certain object categories. For example, objects like tables or skateboards—which exhibit front-back symmetry—are treated differently across datasets. ImageNet3D assumes these objects are always viewed from a canonical “front” (as described in Section A.1 of their paper), whereas in our dataset, they are annotated as having no meaningful orientation using symmetry-based criteria. This definitional mismatch limits the observed improvement when scaling up our training data for evaluation on ImageNet3D.
> > >
> > > However, as shown in our response to W1.1 & Q1.1 from Reviewer 83jQ, when we fine-tune Orient Anything directly on ImageNet3D’s training set, the model quickly adapts to its label definitions and significantly outperforms ImageNet3D-DINOv2-B. This result demonstrates both the transferability and adaptability of our method, indicating its potential as a foundational model for orientation estimation tasks.
> > >
> > > ---
> > >
> > > Once again, we sincerely appreciate your feedback and will clarify these points in the final version of the paper.

---

### Official Review · Reviewer_MxHL · 2025-03-13

**Overall Recommendation:** 3

**Summary:**

The paper introduces Orient Anything, a foundation model for zero-shot object orientation estimation. The key contributions include: 1) Leveraging 3D models and VLMs to annotate front faces, generating 2M synthetic images with orientation labels; 2) Modeling orientation as Gaussian distributions over angles (azimuth, polar, rotation) to improve training stability; 3) Using DINOv2 initialization and data augmentation (random cropping) to bridge domain gaps.; 4) A VQA benchmark revealing VLMs’ limitations in orientation understanding.
Results show state-of-the-art zero-shot performance on real-world datasets (e.g., SUN RGB-D, KITTI) and significant improvements over VLMs (GPT-4o, Gemini) on orientation-related tasks.

**Claims And Evidence:**

Basically yes. 2M images from 55K 3D models across 7,204 categories (vs. 100 in ObjectNet3D). But there's no ablation studies for key components (distribution fitting, augmentation). COCO evaluation’s 8-direction simplification lacks clarity on mapping from 3D angles.

**Essential References Not Discussed:**

N/A

**Experimental Designs Or Analyses:**

​COCO Evaluation uses a simplified 8-direction task, which may not fully reflect 3D orientation. Mapping from predicted angles to directions needs clarification. Manual setting of variances (σ_θ, σ_φ, σ_δ) lacks sensitivity analysis. Impact of random cropping is asserted but not quantified.

**Methods And Evaluation Criteria:**

Synthesizing data via 3D rendering and distribution-based training is sensible. Probability distributions effectively handle angle periodicity. Real-world benchmarks (SUN RGB-D, KITTI) are appropriate but compared unfairly to supervised models.

**Other Comments Or Suggestions:**

N/A

**Other Strengths And Weaknesses:**

Novel use of synthetic data, Ori-Bench benchmark, strong empirical results.

Cons: Sometimes the results are not good in my experiments of running the provided code and checkpoint. Limited ablation studies, unclear real-world evaluation protocol.

**Questions For Authors:**

- How are 3D angles (θ, φ, δ) mapped to 8 horizontal directions in COCO evaluation? Could this simplification misrepresent orientation?
- What is the quantitative effect of data augmentation on synthetic-to-real transfer?
- How were σ_θ, σ_φ, σ_δ chosen? Was sensitivity analysis performed?
- How to extend the proposed method to a video clip, where the object's orientation needs to be estimated in a temporal manner?

**Relation To Broader Scientific Literature:**

Well-situated against 6DoF pose estimation and viewpoint detection. Connects to VLMs’ limitations via Ori-Bench. Missing discussion of recent synthetic-data approaches or self-supervised orientation methods.

**Theoretical Claims:**

No theoretical proofs; methods are empirically validated.

---

> ### Author Rebuttal · Authors · 2025-04-01
>
> Thank you for your positive review for recognizing the significance of our paper and invaluable suggestions. Below we respond to the comments in **Weaknesses (W)** and **Questions (Q)**.
>
> ---
>
> ***W1&Q2&Q3: Limited ablation studies for key components: 1. distribution fitting, 2. augmentation, 3. random cropping, 4. sensitivity for variances (σ_θ, σ_φ, σ_δ).***
>
>
>
> In Table 5 of the current manuscript, we compare **different learning objectives** (continuous regression vs. discrete classification vs. distribution fitting), inference augmentations (box vs. mask), and **training augmentations** (with vs. without random cropping).
>
>
>
> In Figure 5, we analyze the impact of **different selections for (σ_θ, σ_φ, σ_δ)**. Generally, our method is not sensitive to the hyper-parameter.
>
>
>
> We respectfully inquire whether you are referring to other kinds of the ablation study. If so, could you kindly provide more specific settings, and we will make the necessary additions?
>
> ---
>
> ***W2: Clear real-world evaluation protocol.***
>
> Currently, the 6 benchmarks in Table 2 are evaluated on real-world datasets and reported with quantitative 3D orientation error.
>
>
>
> Moreover, we have also tested on the latest and largest 3D orientation benchmark, ImageNet3D [1], which covers 200 object categories. The Acc@30° results are as follows:
>
> |Setting | Model                             | Avg. | Electronics | Furniture | Household | Music | Sports | Vehicles | Work |
> |---| --------------------------------- | ---- | ----------- | --------- | --------- | ----- | ------ | -------- | ---- |
> |Zero-shot| ImageNet3D-ResNet50   | 37.1 | 30.1        | 35.6      | 28.1      | 11.8  | **51.7** | 36.7     | **40.9** |
> || Orient Anything-B     | **48.5** | **61.0** | **66.8** | **37.9** | **27.3** | 25.6 | **70.8** | 33.4 |
> |Fine-tuning| ImageNet3D-ResNet50 | 53.6 | 49.2        | 52.4      | 45.8      | 26.0  | 65.2   | 56.5     | 58.5 |
> || ImageNet3D-DINOv2-B | 64.0 | 75.3        | 47.9      | 32.9      | 23.5  | **74.7** | 38.1     | **64** |
> || Orient Anything-B     | **71.3** | **77.6**    | **89.7**  | **64.4**  | **54.4** | 47.6   | **87.4** | 61.2 |
>
> Note that we couldn't find detailed definitions for the major categories like *Electronics*, *Furniture*, and *Household* in ImageNet3D, so we used GPT to map its 200 categories into the 7 general categories. As a result, we used GPT to map its 200 categories into 7 broader ones. Therefore, the comparison results for each general category may vary, and the average score provides a more meaningful comparison.
>
> [1] Ma W, Zhang G, Liu Q, et al. ImageNet3d: Towards general-purpose object-level 3d understanding[J].  NIPS 2024
>
> ---
>
>
>
> **W3: Missing discussion.**
>
> We will include more related works on synthetic-data approaches and self-supervised orientation methods. We also respectfully inquire if you could provide more specific related works.
>
> ---
>
> ***Q1: Mapping from predicted angles to directions in COCO evaluation.***
>
> As discussed in Section 6.2, the COCO direction focuses only on the horizontal plane (e.g., azimuth angle). Specifically, we simply map the predicted azimuth angle (0-360°) to the 8 directions with 45-degree intervals. For example, 0±22.5° corresponds to the front, 45±22.5° to front-left, 90±22.5° to left, and so on.
>
> ---
>
> ***Q4: Extend to video clip.***
>
> Simply performing per-frame predictions on video data, followed by cross-frame smoothing through simple averaging, can yield relatively consistent and accurate orientation estimation. Some examples are provided in <https://anonymous.4open.science/r/visualization-B728/Video_Cases/>.

---

### Official Review · Reviewer_ev5k · 2025-03-13

**Overall Recommendation:** 3

**Summary:**

This paper proposes Orient Anything, a method that obtains orientation through 3D assets and distilled VLM annotation. Although this paper is somewhat overclaimed, it is pioneering.

**Claims And Evidence:**

Yes

**Essential References Not Discussed:**

None

**Experimental Designs Or Analyses:**

Yes

**Methods And Evaluation Criteria:**

The paper is meaningful, most previous academic research has focused on location or spatial relationships, especially robotic tasks.

But I think this paper is over claims; the author only renders images on 80K 3D objects to generate orientations, which is far fewer than the sample size used in other xxx-Anything works such as Segment Anything, Depth Anything, etc., and even less than the COCO dataset. I am skeptical of its generalizability. In addition, the zero-shot performance in Table 3 appears to be not ideal.

**Other Comments Or Suggestions:**

Recently, SoFar used a similar approach for orientation learning and understanding, and has been filtered and trained on a larger dataset (the full set of Objaverse1.0). I suggest that the authors use similar methods and data to scale up Orient Anything.


SoFar: Language-Grounded Orientation Bridges Spatial Reasoning and Object Manipulation

**Other Strengths And Weaknesses:**

This paper has many experiments and visualizations, and the combination of the model and the LLM has impressed me with its ability to enhance orientation understanding.

**Questions For Authors:**

1. Does Orientation-Anything generalize the position (on the corner) and size (too large or too small) of the object in the image?
2. Many items do not have a clear "front side", such as apples, footballs, light bulbs, etc., how should the orientation of such objects be defined?

**Relation To Broader Scientific Literature:**

The concept of orientation comes from object pose. This article directly obtains the orientation in the camera coordinate system from 2D images and does not depend on any template.

**Theoretical Claims:**

N.A.

---

> ### Author Rebuttal · Authors · 2025-04-01
>
> Thank you for your positive review for recognizing the significance of our paper and invaluable suggestions. Below we respond to the comments in **Weaknesses (W)** and **Questions (Q)**.
>
> ---
>
> ***W1&W2: Scaling up Orient Anything.***
>
> Thank you for your suggestion. The SoFar dataset is really helpful, and we will incorporate it to further scale up.
>
>
> In fact, we are also actively working on expanding our training data to the scale of 10+ million to further improve Orient Anything. Recent breakthroughs in open-source 3D asset generation models (e.g., Hunyuan-3D [1], TRELLIS [2]) have demonstrated impressive results, with outputs now reaching sufficiently high quality. On the other hand, as discussed in Section 7.3 of our manuscript, the initial Orient Anything model can serve as an annotator to robustly label the orientation of 3D assets through multi-view voting. This voting mechanism helps reduce prediction errors, achieving robustness beyond the model's original capabilities.
>
>
>
> The above two observations show the potential to freely increase annotated 3D assets to any scale. In <https://anonymous.4open.science/r/visualization-B728/Synthesized_Assets_and_Voting_Annotation/>, we showcase examples of synthesized 3D assets and the voting-based orientation annotations using Orient Anything. We hope this further observation can address your concerns regarding scaling up.
>
> [1] Zhao Z, Lai Z, Lin Q, et al. Hunyuan3d 2.0: Scaling diffusion models for high resolution textured 3d assets generation[J]. arXiv preprint arXiv:2501.12202, 2025.
>
> [2] Xiang J, Lv Z, Xu S, et al. Structured 3d latents for scalable and versatile 3d generation[J]. arXiv preprint arXiv:2412.01506, 2024.
>
> ---
>
>
>
> ***Q1: Generalization to object position and size.***
>
> In <https://anonymous.4open.science/r/visualization-B728/Corner_TooBig_TooSmall_Case/>, we provide visualizations of objects that are too large, too small, or on the corner. Overall, our model demonstrates generalization to these scenarios.
>
> ---
>
> ***Q2: Items do not have a clear “front side”.***
>
> During annotating, 3D assets lacking a clear "front side" are explicitly labeled as having no front side, as shown in <https://anonymous.4open.science/r/visualization-B728/Training_Samples/>. We identify these cases using two methods: symmetry detection and VLM-based semantic understanding (as illustrated in Figure 3b, "Orientation Annotating").
>
> During training, Orient Anything learns to predict a confidence score indicating the likelihood of an object having a clear front side.
>
> During inference, Orient Anything can predict low confidence scores for objects lacking a clear "front side" to reflect their ambiguous orientation. Actually, the "Judgment" column in Table 3 is the accuracy of judging whether an object has a distinguishable front face.

---

### Decision · Program_Chairs · 2025-05-01

**Decision:**

Accept (poster)

**Comment:**

The authors presented a foundation model for object orientation estimation. The main contributions are in scaling the size of the dataset through the use of 3D models and VLMs, and a loss function. While the reviewers noted that the modeling innovations are weak, e.g., use of DINO pretrained weights, use of small head for output, supervised learning, etc., they agreed that the proposed method produces strong empirical results. The value brought comes largely from laying the ground work in curating the large-scale dataset and training a foundational model for the task.